# AIS data-driven MAAC-Stackelberg multi-ship cooperative collision avoidance algorithm

**Tie Xu**◉❂*, **Tengdong Wang**❂, **Jiansen Zhao**❂, **Qinyou Hu**❂

Merchant Marine College, Shanghai Maritime University, Shanghai, China

❂ These authors contributed equally to this work.
* tiexu@shmtu.edu.cn

## Abstract

Ship collision avoidance has become a focus issue in maritime navigation. Existing methods often struggle to simultaneously meet the hierarchical decision-making requirements of the International Regulations for Preventing Collisions at Sea (COL-REGs), address the dynamic uncertainty of ship risk attitudes, and effectively cope with multi-ship coupling risks. To solve the above problems,this paper proposes an algorithm that combines multi-agent systems with game theory, and integrates ship collision avoidance rules into the reward function design. The algorithm constructs a two-stage framework: the risk attitude perception layer uses a Long Short-Term Memory (LSTM) network to predict the short-term motion states of target ships, and dynamically infers the probability distribution of target ships' risk attitudes through a Bayesian network combined with historical Automatic Identification System (AIS) data and encounter characteristics. The decision-making execution layer integrates Stackelberg game with the Multi-Agent Actor-Critic (MAAC) algorithm, and embeds COLREGs as rigid constraints into the action space to ensure the compliance of the algorithm. Experimental verification is carried out based on historical AIS data and simulation scenarios. The results show that the proposed algorithm has certain advantages in various key indicators,the collision rate, the COLREGs compliance rate, the trajectory smoothness, and the average risk. Statistical significance tests confirm the robustness and superiority of the algorithm. This study provides a reliable technical scheme for ship collision avoidance strategies in multi-ship waters.

## 1. Introduction

With the rapid development of autonomous shipping, maritime traffic safety has become a core focus of the international maritime community. The International Regulations for Preventing Collisions at Sea (COLREGs) provide the fundamental framework for ship collision avoidance, but the complexity of multi-ship encounter scenarios and the uncertainty of maritime environments pose significant challenges to the practical implementation of these regulations [1]. Collision accidents not only

**Data availability statement:** All relevant data are within the paper and its Supporting information files.

**Funding:** The author(s) received no specific funding for this work.

**Competing interests:** The authors have declared that no competing interests exist.

cause enormous economic losses but also lead to severe environmental pollution, making the development of efficient and reliable multi-ship collision avoidance algorithms an urgent demand in ocean engineering [2].

Automatic Identification System (AIS) data, as a key source of maritime traffic information, has been widely used in collision risk assessment, anomaly detection, and ship behavior analysis due to its advantages of high accuracy and real-time performance [3]. Recent studies have demonstrated the potential of AIS data-driven methods in improving the intelligence of collision avoidance systems, such as ship domain estimation [4], risk assessment and navigation decision-making [5]. However, the effective integration of AIS data with advanced decision-making algorithms to solve multi-ship cooperative collision avoidance problems remains a critical research gap.

In the field of multi-agent decision-making, multi-agent actor-critic (MAAC) has emerged as a promising algorithm for cooperative-competitive environments, enabling multiple agents to learn optimal strategies through interactive training [6]. Several studies have applied MAAC and other MARL methods to ship collision avoidance, achieving positive results in two-ship or simple multi-ship scenarios. Nevertheless, these methods often ignore the hierarchical decision-making characteristics of multi-ship encounters, where ships may have different priorities such as give-way ships and stand-on ships as required by COLREGs [7], leading to suboptimal cooperative effects in complex encounter situations.

Game theory provides an effective tool for modeling multi-agent interactive decision-making, and Stackelberg equilibrium as a hierarchical game solution is particularly suitable for scenarios with leader-follower relationships. However, existing game-based methods either focus on two-ship encounters [8] or lack effective integration with MARL algorithms, making it difficult to adapt to the dynamic and cooperative characteristics of multi-ship collision avoidance in real maritime environments. Additionally, although some studies have combined game theory with reinforcement learning, they rarely consider the guidance of AIS data in strategy learning, resulting in poor generalization of the algorithms to actual maritime traffic scenarios [9].

To address the aforementioned limitations. This research introduces a multi-ship cooperative collision avoidance algorithm based on the integration of AIS data and MAAC-Stackelberg frameworks. The core innovations of this work are as follows: (1) Incorporate the preprocessing of AIS data and the extraction of features to deliver precise environmental insights and comprehensive ship state details to enhance the effectiveness of the decision-making system; (2) Integrate MAAC with Stackelberg equilibrium to develop a hierarchical cooperative decision-making model, allowing vessels to modify their strategies based on their COLREGs specified priorities [10]; (3) Design a cooperative reward function considering both collision avoidance safety and navigation efficiency, ensuring the algorithm's compliance with COLREGs and practical engineering applicability.

The remainder of this paper is structured as follows. Section 2 reviews the relevant literature on collision risk assessment, game-theoretic approaches, and multi-agent reinforcement learning in maritime CADM. Section 3 details the methodology, including the risk attitude perception layer based on AIS data, Bayesian reasoning and the

collision avoidance decision execution layer integrating MAAC and Stackelberg game. Section 4 presents the simulation results and analysis, followed by conclusions and future work in Section 5.

## 2. Literature review

### 2.1. Collision risk assessment in maritime navigation

Collision risk assessment is the foundational premise of collision avoidance decision-making, responsible for quantifying the probability and severity of potential collisions to provide decision support for subsequent avoidance maneuvers. In the maritime field, this research has evolved from static geometric indicators to dynamic data-driven models, but key limitations remain in adapting to complex multi-ship and human-in-the-loop scenarios [11].

Traditional collision risk assessment relies heavily on kinematic parameter-based indicators. The Distance to Closest Point of Approach (DCPA) and Time to Closest Point of Approach (TCPA) are the most widely used core metrics, as they can quickly reflect the spatial proximity and temporal urgency of two-ship encounters [12]. However, these indicators are inherently static and fail to capture the dynamic evolution of encounter situations such as sudden speed changes of target ships and the subjective risk preferences of mariners [13].

To address this, the ship domain theory has become a mainstream research direction since its proposal by Fujii (1971) [14]. Early models such as circular and rectangular domains simplified the navigational environment and treated all ships as homogeneous. With the popularization of Automatic Identification System data, recent studies have focused on dynamic and asymmetric ship domains tailored to maritime characteristics. For instance, Silveira et al. (2022) extracted quaternion ship domain parameters from AIS data by considering differences in bow, stern, port and starboard safety distances, which improved the fitting degree with actual navigation behavior [15].

Additionally, data-driven methods have been widely adopted. Wang et al. (2023) used a multi-scale risk estimation model based on AIS data to quantify collision risks in complex waters [16], Guo et al. (2023) proposes the VT-MDM method, which has successfully achieved high-precision multi modal vessel trajectory prediction [17].

Despite these advancements, collision risk assessment in the maritime field still faces three critical limitations. One limitation is that most models such as ship domain and DCPA/TCPA treat risk perception as deterministic, ignoring the variability of mariners' risk attitudes that include conservative, neutral and aggressive types due to experience, education and navigation scenarios [18]. For instance,a conservative mariner may trigger avoidance actions when the target ship is 5 ship lengths away, while an aggressive one may wait until 3 ship lengths, a difference that existing models fail to capture. Another limitation lies in that in multi-ship encounters, the collision risk between any two ships is not independent but mutually coupled, as avoiding Ship A may increase the risk of colliding with Ship B. However, current models either simplify multi-ship scenarios into pairwise interactions or rely on pre-defined rules to reduce complexity [19]. A further limitation is that risk assessment results are often used as independent early warning signals rather than being deeply integrated into subsequent collision avoidance decision-making [20]. This disconnect leads to suboptimal strategies such as overly conservative maneuvers based on high-risk warnings that compromise navigation efficiency.

### 2.2. Game-theoretic approaches in maritime CADM

Game theory provides a rigorous mathematical framework for modeling interactive decision-making among multiple vessels, where each ship's strategy depends on the expected behaviors of others. In the maritime field, game-theoretic approaches have been extensively probed to resolve collision avoidance conflicts, with particular emphasis on Nash equilibrium and Stackelberg equilibrium as core analytical paradigms but significant gaps remain in adapting to incomplete information and COLREGs compliance.

Nash equilibrium is a prevailing game-theoretic model applied to maritime CADM, assuming simultaneous and independent decision-making by all ships. Wan et al. (2025) proposes a cooperative collision avoidance framework integrating game theory and intelligent optimization [21].

The Stackelberg equilibrium,alternatively denoted as the leader-follower paradigm,has garnered growing scholarly traction in recent years.This model assumes that the leader,which comprises stand-on ships and large-tonnage vessels,-makes decisions first,while the follower,which consists of give-way ships and small vessels, responds optimally.Additionally,probabilistic game theory has been explored to address uncertainty.A probabilistic Stackelberg game is introduced into autonomy using a leader–follower structure,aiming to compute equilibrium strategies under perceptual uncertainty.

Game-theoretic approaches in maritime CADM have key shortcomings. One limitation is that most models assume perfect knowledge of other ships' intentions, risk attitudes, and maneuver capabilities, but in reality, AIS data may be noisy, delayed, or lost, and human operated ships exhibit unpredictable behaviors, leading to incomplete information. Another limitation is that existing models in the maritime field often use fixed rules such as tonnage and COLREGs status to identify leaders and followers, failing to dynamically adjust based on real-time encounter evolution; for instance, a sudden speed reduction by the leader may require role switching. A further limitation is that the utility functions in game models are typically designed based on universal safety and efficiency criteria, ignoring differences in risk attitudes between ships. For instance, an aggressive follower may not respond to a leader's maneuver as expected, leading to strategy failure [22].

### 2.3. Multi-agent reinforcement learning in maritime CADM

MARL enables ships to learn nearly optimal collision avoidance strategies through interaction with the environment and other agents,making it suitable for dynamic multi-ship scenarios. The MAAC algorithm,distinguished by its architecture of centralized training and decentralized execution, has emerged as a excellent area of research within maritime CADM. However, its implementation continues to encounter difficulties with adherence to rules and scalability.

Multi-Agent Reinforcement Learning algorithms such as Multi-Agent Deep Deterministic Policy Gradient (MADDPG), Multi-Agent Actor-Critic with Attention (MAAC) and Q-Mixing Network (QMIX) have been widely explored in maritime collision avoidance. For example,Wang et al.(2025) proposes a communication-integrated multi-agent deep deterministic policy gradient algorithm to address cooperative control issues for Unmanned Surface Vehicles (USV) in USV-based multi-agent systems under partial observability and non-stationarity. The algorithm is verified via cooperative navigation and collision avoidance tasks, outperforms traditional single-agent methods and enables effective coordination by establishing communication protocols and sharing observations to compensate for missing information [23]. Zhu et al. (2024) proposes a multi-ship autonomous collision avoidance decision-making algorithm based on MER-D3QN to enables all ships to complete collision avoidance tasks safely and efficiently [24].

MARL in maritime CADM has three limitations. One limitation is that although some studies incorporate COLREGs into reward functions, the rules are often simplified, for example by adopting binary rewards that distinguish between compliance and non-compliance, rather than being deeply embedded in the learning process. This leads to strategies that may violate detailed provisions such as those related to the timing and amplitude of maneuvers in complex scenarios.Another limitation is that MARL's training efficiency and strategy performance depend heavily on reasonable state partitioning, but existing multi-ship partitioning methods fail to account for dynamic risk attitudes and coupled risks, leading to unstable learning in dense traffic. A further limitation is that MARL relies on large-scale AIS data for training, but real-world data often contains noise, gaps, and outliers. Current models lack effective preprocessing and adaptation mechanisms, resulting in degraded performance in practical applications [25].

### 2.4. Contributions and innovations of this study

To address these gaps,this study proposes a novel collision avoidance strategy combining the MAAC algorithm and the Stackelberg game based on AIS data analysis, with the following key contributions and innovations. Probabilistic dynamic modeling of risk attitudes for maritime scenarios. By analyzing historical AIS data to extract collision avoidance behaviors, we establish a prior distribution of risk attitudes,and use Bayesian reasoning to dynamically update the probability

distribution of target ships' risk attitudes in real time. This overcomes the limitation of static risk attitude assumptions. Stackelberg game with incomplete information tailored to maritime characteristics. We design a dynamic leader-follower identification mechanism based on AIS-derived kinematic features including distance to conflict point and maneuverability as well as COLREGs rules, and construct a utility function that integrates risk attitude probabilities. This addresses the problem of static role partitioning and complete information assumptions in traditional maritime game models, ensuring hierarchical decision-making consistency with practical navigation. MAAC algorithm with deep COLREGs integration and multi-ship coupling adaptation. We embed detailed COLREGs provisions including maneuver timing and direction into the reward function, and optimize the MAAC's centralized critic network to capture multi-ship coupled risks. This improvement in rule adherence and decision-making rationality of the acquired strategies addresses the issues of superficial rule integration and limited adaptability to multiple ships found in current MAAC-based maritime CADM systems. We unified multi-ship encounter state partitioning framework. We propose a two-stage partitioning method that first classifies basic encounter states including head-on, crossing and overtaking based on AIS kinematic data, then refines sub-states according to risk attitude probabilities and coupled risk levels. This provides a scalable foundation for multi-ship decision-making,bridging the gap between two-ship and multi-ship state modeling.

## 3. Methodology

This segment offers a comprehensive overview of the technical execution of the AIS data-driven MAAC-Stackelberg multi-ship cooperative collision avoidance algorithm. Beginning with data preprocessing, progressing to state identification,-continuing through prior analysis and risk assessment, then advancing to prediction inference,and culminating in decision optimization. Fig 1 gives the methodology framework and more details are discussed below.

### 3.1. AIS data preprocessing

AIS data serves as the core input for the entire framework, including dynamic parameters position $(x,y)$, speed $v$, heading $\psi$, rate of turn $r$, static parameters ship length $L$, width $B$, ship type $T$, and timestamp $t$. To eliminate noise and ensure data reliability, a three-stage preprocessing pipeline is implemented, referring to mature AIS data processing protocols in maritime research.

Abnormal data such as signal drift, sudden jumps and missing values are identified using the $3\sigma$ principle and kinematic constraints, consistent with the data quality control method. For dynamic parameters including $v, \psi$ and $r$, the anomaly detection criterion is defined as follows:

$$|p_i - \mu_p| > 3\sigma_p \quad \text{or} \quad |p_i - p_{i-1}| > \Delta p_{\max}, \tag{1}$$

where $\mu_p$ and $\sigma_p$ are the mean and standard deviation of parameter $p$ in a sliding window with a window size of 30s, and $\Delta p_{\max}$ is the maximum allowable change rate such as $\Delta v_{\max} = 2\,\text{kn/s}$ and $\Delta \psi_{\max} = 5°/\text{s}$, which is determined based on ship maneuverability constraints.

Missing values are supplemented using cubic spline interpolation to ensure trajectory smoothness:

$$p(t) = a_0 + a_1 t + a_2 t^2 + a_3 t^3, \tag{2}$$

where $a_0, a_1, a_2, a_3$ are interpolation coefficients solved by minimizing the second derivative of the trajectory.

### 3.2. Trajectory reconstruction

To unify the time interval of AIS data with the original interval ranging from 1s to 10s, the trajectory is resampled at $\Delta t = 1$ s using linear interpolation for dynamic parameters. The reconstructed trajectory of ship $i$ is represented as follows:

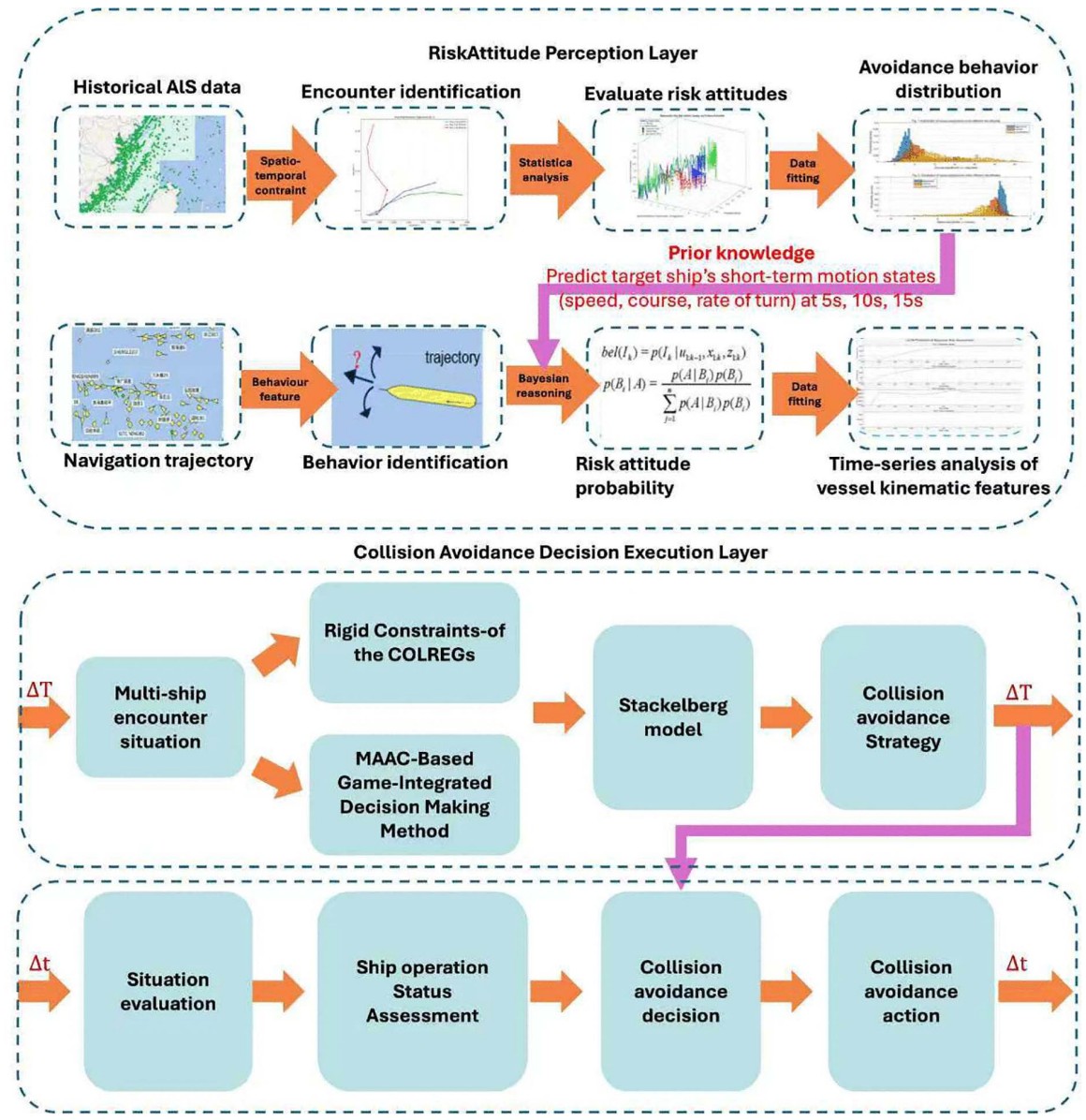

**Fig 1. Framework of the ship cooperative collision avoidance algorithm.** The overall architecture of the proposed cooperative collision-avoidance system.

$$\mathcal{T}_i = \{(x_i(t), y_i(t), v_i(t), \psi_i(t), r_i(t)) \mid t \in [t_0, t_N]\}, \tag{3}$$

where $t = t_0 + k\Delta t, k = 0, 1, ..., N$.

To facilitate distance computation, the initial WGS-84 geodetic coordinates denoted by latitude $\phi$ and longitude $\lambda$ are transformed into a local Cartesian coordinate system $(x, y)$ through the application of a simplified UTM transverse Mercator approximation. This approximation is appropriate for relatively small study regions, generally those that do not exceed 100 km. The transformation process is formulated as follows:

$$\begin{cases} x = (\lambda - \lambda_0) \cdot \dfrac{\pi R}{180} \cos \phi_0, \\ y = (\phi - \phi_0) \cdot \dfrac{\pi R}{180}, \end{cases} \tag{4}$$

where $\phi$ and $\lambda$ are the geodetic latitude and longitude in degrees of the point to be converted. $\phi_0$ and $\lambda_0$ denote the latitude and longitude of the local coordinate origin usually chosen as the centroid of the study region. $R = 6378$ km is the WGS-84 reference ellipsoid semi-major axis. The factor $\pi/180$ converts angular values from degrees to radians for trigonometric consistency. This local planar approximation replaces spherical distance calculations with Euclidean computations, reducing computational complexity in trajectory analysis and spatial positioning.

### 3.3. Two-ship encounter identification

In accordance with COLREGs and standard maritime navigation practices, two-ship encounters are detected by fusing kinematic indicators including DCPA and TCPA and spatial criteria including ship domain intrusion.

For ship $i$ (own ship, OS) and ship $j$ (target ship, TS), the relative motion vector is $(\Delta v_x, \Delta v_y) = (v_j \cos \psi_j - v_i \cos \psi_i, v_j \sin \psi_j - v_i \sin \psi_i)$, and the initial relative position is $(\Delta x_0, \Delta y_0) = (x_j(t_0) - x_i(t_0), y_j(t_0) - y_i(t_0))$. The TCPA and DCPA are calculated using the classic kinematic model:

$$TCPA = -\frac{\Delta x_0 \Delta v_x + \Delta y_0 \Delta v_y}{\Delta v_x^2 + \Delta v_y^2} \quad , \tag{5}$$

$$DCPA = \sqrt{(\Delta x_0 + \Delta v_x \cdot TCPA)^2 + (\Delta y_0 + \Delta v_y \cdot TCPA)^2}. \tag{6}$$

An encounter is identified if TCPA > 0 (future encounter) and DCPA < $D_{safe}$ (unsafe proximity), where $D_{safe} = 3L_{max}$ ($L_{max} = \max(L_i, L_j)$). Three canonical encounter states are classified using the relative bearing $\beta$ and relative speed ratio $\gamma = v_j/v_i$, consistent with COLREGs rules:

$$\begin{cases} \text{Head-on} & \beta \in [355°, 5°] \cap \Delta\psi \in [165°, 195°] \cap \gamma \in [0.8, 1.2], \\ \text{Overtaking} & \beta \in [112.5°, 247.5°] \cap |\Delta\psi| \leq 30° \cap (\gamma > 1.0 \vee \gamma < 0.8), \\ \text{Crossing} & \beta \in (15°, 112.5°) \cup (247.5°, 345°) \cap \Delta\psi \in (30°, 165°) \cup (195°, 330°), \\ \text{Safe Encounter} & \text{otherwise.} \end{cases} \tag{7}$$

To further refine encounter patterns for complex multi-ship scenarios, the azimuth-based sub-classification and encounter pattern mapping are established as shown in Table 1 and Table 2.

Traditional models for two-ship encounter situations rely on pairwise DCPA and TCPA calculations and fixed relative bearing rules to identify and classify head-on, crossing, and overtaking encounter states. But in a real environment, ship roles can change dynamically. For instance, a target ship (TS) in one encounter may act as an own ship (OS) in another ship pair's encounter. Meanwhile, risk attitudes are heterogeneous among different target ships, which may exhibit conservative, neutral, or aggressive navigation behaviors. This can further influence collision avoidance decision-making. To remediate these aforementioned challenges, we use Table 1 and Table 2 functioning as the linchpin tools underpinning state structuring. The Table 1 and Table 2 adopt the azimuth Az instead of $\beta$, because Az as a global angle is more stable in complex multi-ship scenarios and will not fluctuate frequently with the turning of the own ship, making it more suitable for multi-ship relationship modeling. Based on this, Table 1 discretizes the continuous azimuth Az into 8 sub-classes to realize the coding of orientation information,

**Table 1. Ship encounter azimuth mapping characteristics.**

| Azimuth | Sub-classification | Azimuth range/° |
|---|---|---|
| A | A1 | $2 \leq Az < 5$ |
| | A2 | $355 \leq Az < 360$ or $0 \leq Az < 2$ |
| B | B1 | $5 \leq Az < 44$ |
| | B2 | $44 \leq Az < 108$ |
| C | C1 | $108 \leq Az < 203$ |
| | C2 | $203 \leq Az < 252$ |
| D | D1 | $252 \leq Az < 306$ |
| | D2 | $306 \leq Az < 355$ |

Sub-classification (A1–D2) and corresponding azimuth ranges used for multi-ship encounter state partitioning.

**Table 2. Classification of ship encounter patterns.**

| No. | Encounter pattern | Encounter situation |
|---|---|---|
| 1 | A1-A1 | Head-on |
| 2 | A1-A2 | Head-on |
| 3 | A2-A2 | Head-on |
| 4 | B1-B1 | Crossing-(Head-on solution) |
| 5 | D2-D2 | Crossing-(Head-on solution) |
| 6 | A1-B1 | Crossing-(Head-on solution) |
| 7 | A1-B2 | Crossing |
| 8 | A1-D1 | Crossing |
| 9 | A1-D2 | Crossing |
| 10 | A2-B1 | Crossing-(Head-on solution) |
| 11 | A2-B2 | Crossing |
| 12 | A2-D1 | Crossing |
| 13 | A2-D2 | Crossing-(Head-on solution) |
| 14 | B1-D1 | Crossing |
| 15 | B1-D2 | Crossing |
| 16 | B2-D1 | Crossing |
| 17 | B2-D2 | Crossing |
| 18 | A1-C1 | Overtaking |
| 19 | A1-C2 | Overtaking |
| 20 | A2-C1 | Overtaking |
| 21 | A2-C2 | Overtaking |
| 22 | B1-C2 | Overtaking |
| 23 | B2-C2 | Overtaking |
| 24 | C1-D2 | Overtaking |
| 25 | Others | – |

25 pairwise patterns mapped to Head-on, Crossing, or Overtaking situations for coupled-risk estimation.

laying a foundation for the refined classification of encounter patterns in Table 2. For an OS and $n$ TSs where $j$ takes values from 1 to $n$, the relative azimuth $Az_{ij}$ is defined as the angle of TS $j$ relative to OS $i$'s navigation coordinate system, and it is calculated for each ship pair to form an initial multi-ship azimuth set $\{A_{i1}, A_{i2}, \ldots, A_{in}\}$.

Table 1 resolves the lack of granularity in two-ship state classification by subdividing azimuths into 8 sub-classifications including A1/A2,B1/B2,C1/C2 and D1/D2 based on maritime kinematic rules. For each $Az_{ij}$ in the multi-ship azimuth set, match $Az_{ij}$ to the Azimuth range in Table 1 to determine the corresponding sub-classification. For instance, when $Az_{i1} = 3°$, it is classified as A1. When $Az_{i2} = 50°$, it is classified as B2. This step converts unstructured multi-ship azimuth data into structured sub-classification labels, forming a multi-ship sub-classification set $\{S_{i1}, S_{i2}, \ldots, S_{in}\}$ where $S_{ij}$ represents the sub-classification of TS $j$ relative to OS $i$.

Table 2 expands pairwise encounter patterns to multi-ship scenarios by establishing a mapping between sub-classification pairs and encounter situations. For each TS$j$, construct the sub-classification pair $S_{ij}$, $S_{ji}$ where $S_{ji}$ is the sub-classification of OS $i$ relative to TS $j$ and calculated via the same azimuth matching method as the previous step; match the sub-classification pair $S_{ij}$, $S_{ji}$ to the encounter pattern column in Table 2 to determine the pairwise encounter situation, for example, the sub-classification pair A1-A1 maps to Head-on and B1-D2 maps to Crossing. For multi-ship coupled scenarios such as TS1 and TS2 both forming Crossing situations with OS, refer to the "Crossing-(Head-on solution)" situation in Table 2 and integrate clustering-based multi-ship grouping to classify TSs with overlapping encounter domains into the same group, and prioritize collision avoidance operations for groups with smaller TCPA. This step forms a multi-ship encounter pattern set $\{P_{i1}, P_{i2}, \ldots, P_{in}\}$ where $P_{ij}$ represents the refined encounter situation of TS $j$ relative to OS $i$, achieving standardized description of multi-ship states.

After completing the determination of ship encounter situations,we can conduct prior analysis on AIS data in combination with the identified situation characteristics to provide targeted data support for subsequent research.

### 3.4. Prior analysis of AIS data

Prior analysis of historical AIS data aims to extract statistical characteristics of collision avoidance behaviors, laying the foundation for subsequent risk attitude modeling and algorithm training [26].

A collision avoidance behavior is defined as a maneuver where the ship's turning angle $\Delta\psi = |\psi(t + \Delta t) - \psi(t)| > 5°$ or speed change $\Delta v = |v(t + \Delta t) - v(t)| > 1$ kn during an encounter. The probability distribution of core avoidance parameters including turning angle $\Delta\psi$ and avoidance timing $t_{\text{lead}}$ is fitted using the Weibull distribution which is suitable for non-negative continuous variables in maritime engineering, with $t_{\text{lead}}$ defined as the difference between $t_0$ and $t_{\text{start}}$ and $t_0$ being the time when DCPA equals $D_{\text{safe}}$:

$$f(x; \kappa, \lambda) = \frac{\kappa}{\lambda} \left(\frac{x}{\lambda}\right)^{\kappa-1} e^{-(x/\lambda)^{\kappa}},$$

(8)

where $\kappa$ is a shape parameter,reflecting the concentration degree of collision avoidance behavior. When $\kappa > 1$, collision avoidance actions are concentrated, and when $\kappa < 1$, they are dispersed. $\lambda$ is the scale parameter, corresponding to the characteristic value of the collision avoidance action initiation time, which is obtained through maximum likelihood estimation from historical AIS trajectories. The parameter $\lambda$ is set to 180 s.

$$(\hat{\kappa}, \hat{\lambda}) = \arg \max_{\kappa>0, \lambda>0} \sum_{k=1}^{M} \ln \left(\frac{\kappa}{\lambda} \left(\frac{x_k}{\lambda}\right)^{\kappa-1} e^{-(x_k/\lambda)^{\kappa}}\right),$$

(9)

where $M$ is the number of historical avoidance behavior samples. An eccentric elliptical ship domain is adopted widely used in modern maritime collision risk assessment, with the OS at the left rear of the ellipse center. The domain boundary is:

$$\left(\frac{x'}{aL}\right)^2 + \left(\frac{y'}{bL}\right)^2 = 1, \tag{10}$$

where $(x', y')$ is the relative coordinate of TS relative to OS rotated by OS's heading $\psi_i$, $a$ takes the value 3.5 and serves as the long-axis coefficient corresponding to the bow direction, $b$ takes the value 1.5 and serves as the short-axis coefficient corresponding to the port-starboard direction,and $L$ is OS's length [27]. According to the boundary constraint of the elliptical ship domain established above, the spatial risk $R_s$ is defined by the domain violation degree $d_{dv}$, which refers to the ratio of TS's intrusion depth to domain semi-axis:

$$R_s = \begin{cases} 0 & d_{dv} \leq 0, \\ 1 - e^{-k_s d_{dv}} & d_{dv} > 0, \end{cases} \tag{11}$$

where $k_s = 2.3$ is the decay coefficient calibrated via AIS data.

On the temporal dimension,the temporal risk $R_t$ is constructed to characterize the time urgency of collision risk,which is calculated based on TCPA normalized by the critical time $t_c$. The critical time $t_c$ takes the value of 600s, equivalent to 10 min, and the calculation formula is given as:

$$R_t = \begin{cases} 1 & \text{TCPA} \leq 0, \\ \left(1 - \frac{\text{TCPA}}{t_c}\right)^2 & 0 < \text{TCPA} \leq t_c, \\ 0 & \text{TCPA} > t_c. \end{cases} \tag{12}$$

To achieve a comprehensive and quantitative evaluation of ship collision risk from both spatial and temporal dimensions, the comprehensive collision risk index is obtained by weighted fusion of $R_s$ and $R_t$. The weights used in the fusion process are determined via the Analytic Hierarchy Process (AHP), and the fusion formula is expressed as:

$$R = \omega_s R_s + \omega_t R_t, \tag{13}$$

where $\omega_s = 0.52$ is the spatial risk weight, and $\omega_t = 0.48$ is the temporal risk weight. Both of them are determined by the Analytic Hierarchy Process (AHP).

### 3.5 Feature engineering

To support subsequent prediction and inference models, 22 dimensional features are extracted,divided into three categories referring to the feature selection framework for maritime collision avoidance. Dynamic features include $v_i, \psi_i, r_i, v_j, \psi_j, r_j$ which are ship motion parameters. Encounter features include DCPA, TCPA, $\beta, \gamma, d_{dv}, R$ which are calculated through the corresponding relational expressions. Static features include $L_i, B_i, L_j, B_j, T_i, T_j$. Features are standardized using Z-score normalization to eliminate scale differences, a method common for MARL algorithms:

$$x_{\text{norm}} = \frac{x - \mu_x}{\sigma_x}, \tag{14}$$

where $\mu_x$ and $\sigma_x$ are the mean and standard deviation of feature $x$ in the training set.Based on prior AIS data, key scenario parameters including initial relative position $(\Delta x_0, \Delta y_0)$, speed $v_i, v_j$ and heading $\psi_i, \psi_j$ are sampled from their empirical distributions. For each sample, DCPA and TCPA are calculated through corresponding relational expressions to determine valid encounters. Where TCPA is greater than 0 and DCPA is less than $D_{\text{safe}}$.

For each valid scenario, TS's avoidance behavior is simulated using the prior distribution of $\Delta\psi$ and $t_{\text{lead}}$. The maneuver process is modeled via the Nomoto ship motion equation which is classic for maritime maneuverability:

$$\dot{r} + \frac{1}{T}r = \frac{K}{T}\delta,$$
(15)

where $r$ is the rate of turn, $\delta$ is the rudder angle, $K$ is the rudder gain and $T$ is the time constant which is calibrated by ship type from AIS data. The simulated avoidance trajectory is:

$$\mathcal{T}_j^s(t) = \left( x_j(t_0) + \int_{t_0}^t v_j(\tau)\cos\psi_j(\tau)d\tau, \ y_j(t_0) + \int_{t_0}^t v_j(\tau)\sin\psi_j(\tau)d\tau \right),$$
(16)

where $\psi_j(t) = \psi_j(t_0) + \int_{t_0}^t r_j(\tau)d\tau$.

Simulated avoidance behaviors are aggregated to obtain the avoidance distribution $\mathcal{D} = \{f_{\Delta\psi}, f_{\Delta v}, f_{t_{\text{lead}}}\}$. The 95% confidence interval of each distribution is:

$$CI_{95\%} = \left[ \hat{\mu} - 1.96 \cdot \frac{\hat{\sigma}}{\sqrt{N_s}}, \ \hat{\mu} + 1.96 \cdot \frac{\hat{\sigma}}{\sqrt{N_s}} \right].$$
(17)

## 3.6. LSTM-BN hybrid framework for target ship state

To address the uncertainty of target ship motion and the ambiguity of risk attitude in maritime encounter scenarios, a hybrid framework integrating Long Short-Term Memory (LSTM) and Bayesian Network (BN) is proposed. This framework first leverages LSTM to capture temporal dependencies in AIS data for predicting future motion states, and then fuses the predicted states with static or dynamic encounter features as inputs to BN for probabilistic inference of target ship risk attitude. The synergistic integration of LSTM's temporal prediction capability and BN's uncertainty reasoning advantage provides reliable state support and risk prior information for subsequent cooperative collision avoidance decision-making, which is consistent with the hybrid intelligent reasoning paradigm in maritime safety research. The specific process is shown in the Fig 2.

For prediction, given the time-series nature of AIS data including speed $v_j$, course $\psi_j$ and rate of turn $r_j$. An LSTM network with fully-connected preprocessing and postprocessing layers is designed to predict the target ship's motion states at future key time steps of 5 s, 10 s and 15 s. The input of the LSTM network is a 6-step historical AIS feature sequence $\mathcal{S}_{\text{hist}}$ consisting of feature vectors $\mathbf{f}_t$ from $t = 0$ to $t = 5$, where each time-step feature vector $\mathbf{f}_t$ is a 3-dimensional vector in $\mathbb{R}^3$ with elements $v_j(t), \psi_j(t)$ and $r_j(t)$. To enhance feature expression ability, a fully-connected preprocessing layer maps each 3 dimensional time step feature to a high dimensional space:

$$\mathbf{f}_t^{(\text{high})} = \text{ReLU}(W_1\mathbf{f}_t + b_1),$$
(18)

where $W_1 \in \mathbb{R}^{128 \times 3}$ and $b_1 \in \mathbb{R}^{128}$ are the weight matrix and bias vector of the preprocessing layer respectively. The processed sequence $\mathcal{S}_{\text{pre}} = \{\mathbf{f}_t^{(\text{high})}\}_{t=0}^5 \in \mathbb{R}^{6 \times 128}$ is fed into a two-layer LSTM network to model temporal dependencies:

$$\mathbf{h}_t, (\mathbf{h}_n, \mathbf{c}_n) = \text{LSTM}(\mathbf{f}_t^{(\text{high})}, (\mathbf{h}_{t-1}, \mathbf{c}_{t-1})),$$
(19)

where $\mathbf{h}_t \in \mathbb{R}^{128}$ is the hidden state at time $t$, $\mathbf{h}_n$ is the final hidden state and $\mathbf{c}_n$ is the final hidden state and cell state of the LSTM network respectively. A dropout layer with a dropout rate of 0.2 is added between the two LSTM layers to mitigate overfitting. The final hidden state $\mathbf{h}_n$ is input to a fully-connected postprocessing layer for motion state prediction:

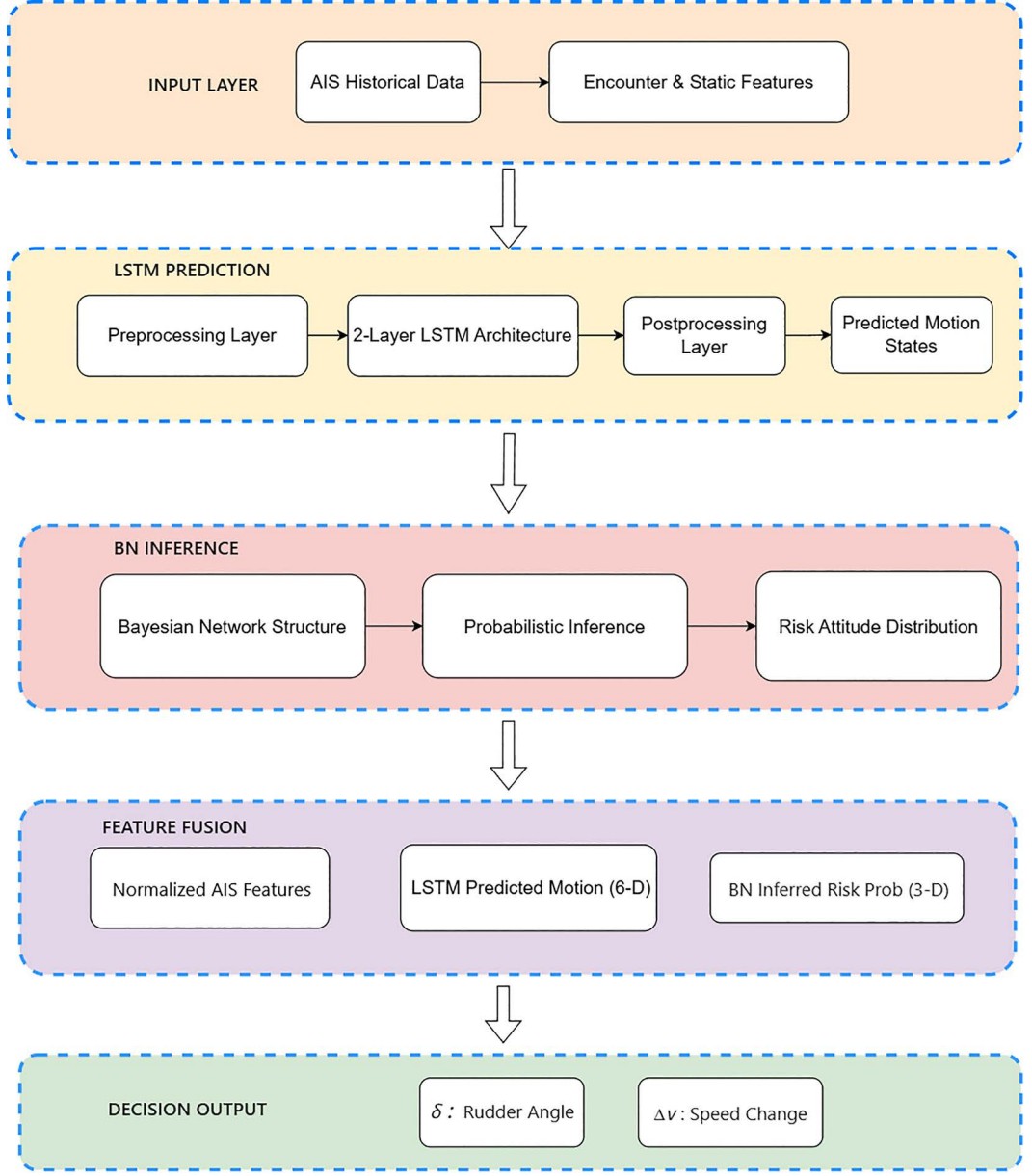

**Fig 2. LSTM-BN framework.** The overall architecture of the proposed cooperative collision-avoidance system.

$$\hat{S}_{\text{fut}} = \text{Linear}(W_3 \cdot \text{ReLU}(W_2 \mathbf{h}_n + b_2) + b_3), \tag{20}$$

where $W_2 \in \mathbb{R}^{64 \times 128}$, $b_2 \in \mathbb{R}^{64}$, $W_3 \in \mathbb{R}^{6 \times 64}$, $b_3 \in \mathbb{R}^6$ are trainable parameters. The output $\hat{S}_{\text{fut}} = [\hat{v}_j(5), \hat{\psi}_j(5), \hat{r}_j(5), \hat{v}_j(10), \hat{\psi}_j(10), \hat{r}_j(10)]^T \in \mathbb{R}^6$ represents the predicted motion states of the target ship at 5s, 10s, and 15s in the future. The LSTM network is trained with the Adam optimizer (learning rate $\eta = 10^{-4}$) and Mean Squared Error loss function, as recommended for maritime motion prediction tasks [28].

The BN is constructed to infer the target ship's risk attitude by fusing LSTM-predicted motion states and original AIS encounter features. The BN is formally defined as BN = (V, E, P). Node set V includes 1 target node and 10 feature nodes. The feature nodes are divided into two categories. LSTM-predicted motion features $\hat{F}_{motion} = \{\hat{v}_j, \hat{\psi}_j, \hat{r}_j\}$ adopted from the 5 s predicted results for real-time decision-making. Original AIS encounter or static features $F_{encounter} = \{DCPA, TCPA, \Delta\psi, t_{lead}, R, L_j, T_j\}$, where DCPA and TCPA characterize collision risk, $\Delta\psi$ is the relative course. $t_{lead}$ is the lead time. $R$ is the encounter range. $L_j$ is the target ship length. $T_j$ is the ship type. Edge set $\mathcal{E}$ directed edges from all 10 feature nodes to the target node $A$, indicating the causal relationship between feature variables and risk attitude. Conditional Probability $Table(CPT)P(A|F_1, ..., F_{10})$ calibrated using historical AIS trajectories, where the prior probability of risk attitude is determined by statistical analysis of collision avoidance behaviors. According to the statistical results, $P(A_1) : P(A_2) : P(A_3) = 5 : 8 : 7$. Based on Bayes' theorem, the posterior probability of the target ship belonging to risk attitude $A_k$ ($k = 1,2,3$) is calculated as:

$$P(A_k|F_1, ..., F_{10}) = \frac{P(F_1, ..., F_{10}|A_k)P(A_k)}{\sum_{m=1}^{3} P(F_1, ..., F_{10}|A_m)P(A_m)},$$

(21)

where $P(F_1, ..., F_{10}|A_k)$ denotes the likelihood function of feature variables under risk attitude $A_k$. Considering the continuity of maritime features, this likelihood function follows a multivariate Gaussian distribution [29]:

$$P(F|A_k) = \frac{1}{(2\pi)^{d/2}|\Sigma_k|^{1/2}} \exp\left(-\frac{1}{2}(F - \mu_k)^T\Sigma_k^{-1}(F - \mu_k)\right),$$

(22)

where $d = 10$ is the number of feature nodes, $F = [F_1, ..., F_{10}]^T$ is the feature vector and $\mu_k \in \mathbb{R}^{10}, \Sigma_k \in \mathbb{R}^{10 \times 10}$ are the mean vector and covariance matrix of features calibrated from historical data under risk attitude $A_k$ respectively.

To construct a comprehensive decision feature set for the subsequent MAAC-Stackelberg collision avoidance algorithm, three types of features are fused. We normalized original AIS features $\mathcal{F}_{norm}$. LSTM-predicted motion states $\hat{S}_{fut}$ including 6 dimensions, $\hat{v}_j(5), \hat{\psi}_j(5), \hat{r}_j(5), \hat{v}_j(10), \hat{\psi}_j(10), \hat{r}_j(10)$. The BN-inferred risk attitude probabilities are given by $\mathcal{P}_A = \{P(A_1), P(A_2), P(A_3)\}$. The fused decision feature set is defined as:

$$\mathcal{F}_{dec} = \mathcal{F}_{norm} \cup \hat{S}_{fut} \cup \mathcal{P}_A,$$

(23)

resulting in a 28-dimensional feature vector. To eliminate redundant information and enhance decision relevance, feature selection is performed using Mutual Information. Specifically, features with mutual information between the feature and optimal collision avoidance actions including rudder angle $\delta$ and speed change $\Delta v$ greater than 0.1 are retained:

$$\mathcal{F}_{dec}^* = \{F \in \mathcal{F}_{dec} \mid MI(F, [\delta, \Delta v]^T) > 0.1\}.$$

(24)

This selection criterion ensures that the final feature set is both compact and discriminative, laying a foundation for efficient and accurate collision avoidance decision-making.

### 3.7. The framework of COLREGs embedded MAAC-Stackelberg

The decision module integrates Stackelberg game which focuses on hierarchical decision and MAAC which emphasizes multi-agent collaboration with rigid COLREGs embedding, combining game-theoretic reasoning and MARL adaptive learning.

Key COLREGs rules are converted to mathematical constraints for decision variables ($\delta, \Delta v$):

- Rule 13 (Overtaking): Overtaking ship $\delta \geq 10°$ (starboard turn), $\Delta v \leq -1$ kn (reduce speed);

- Rule 14 (Head-on): $\delta \geq 10°$ (starboard turn), $\Delta v \geq -2$ kn (no excessive deceleration);

- Rule 15 (Crossing): TS on OS's port side $\delta \geq 15°$ (starboard turn); TS on starboard side $\delta = 0°$ (maintain course).

These constraints are embedded into the MAAC action space:

$$\mathcal{A} = \{(\delta, \Delta v) \mid \delta \in [-\delta_{\max}, \delta_{\max}], \ \Delta v \in [-\Delta v_{\max}, \Delta v_{\max}], \tag{25}$$

where $\delta_{\max}$ takes the value of 30° and $\Delta v_{\max}$ takes the value of 5 kn, with both being consistent with ship maneuverability limits.

In the leader-follower identification, leader L possesses a higher priority level.

$$P_{\text{priority}} = \omega_1 T + \omega_2 L - \omega_3 \text{DCPA}, \tag{26}$$

where $\omega_1 = 0.4, \omega_2 = 0.3, \omega_3 = 0.3$. Follower $F$ is the other ship.

The leader's utility aims to maximize safety and efficiency:

$$U_L = \alpha(v_L + \Delta v_L) - \beta R_L - \gamma |\delta_L|, \tag{27}$$

The follower's utility is to respond to the leader's action:

$$U_F = \alpha(v_F + \Delta v_F) - \beta R_F - \gamma |\delta_F| - \delta |\text{Action}_F - \text{Action}_L^*|, \tag{28}$$

where $\text{Action}_L^*$ is the leader's optimal action. $\alpha = 0.6, \beta = 0.3, \gamma = 0.05, \delta = 0.05$ which are calibrated via simulation.

Solved via backward induction:

$$\text{Action}_F^* = \arg \max_{\text{Action}_F \in \mathcal{A}} U_F(\text{Action}_F | \text{Action}_L), \tag{29}$$

$$\text{Action}_L^* = \arg \max_{\text{Action}_L \in \mathcal{A}} U_L(\text{Action}_L | \text{Action}_F^*). \tag{30}$$

Under the Centralized Training-Decentralized Execution (CTDE) framework, the MAAC algorithm improves the coordination efficiency of multi-agents through the attention mechanism,but suffers from defects such as slow convergence and multi-solution uncertainty. To address these issues, the Stackelberg Equilibrium (SE) is incorporated into the objective function as a regularization term. In the training phase, the central critic utilizes global information to suppress strategy mutual interference, narrow the exploration space,and guide the convergence to a unique subgame perfect equilibrium through the sequential constraints of SE. In the execution phase, the leader and follower agents make independent decisions based on local observations without communication. This design not only retains the engineering feasibility of the CTDE framework, but also solves the coordination and convergence problems of MAAC through the inductive bias of leader-follower games, which is suitable for large-scale hierarchical decision-making scenarios such as power grid regulation and traffic control.

Centralized Critic:

$$Q_\theta(s, a_1, a_2) = \mathbb{E}[r + \gamma Q_{\theta'}(s', a_1', a_2') | s, a_1, a_2], \tag{31}$$

Decentralized Actors generate ship-specific actions for real-time execution:

$$a_i = \pi_{\phi_i}(s_i) + \epsilon \quad (\epsilon \sim \mathcal{N}(0, \sigma^2)),$$

(32)

where $\sigma = 0.1$ is the exploration noise coefficient. The selection of this parameter is synergistically matched with the core hyperparameters of the algorithm,including the learning rate of the Adam optimizer and the regularization coefficient of the Stackelberg equilibrium,which can effectively guarantee the comprehensive performance of the algorithm in terms of collision avoidance safety, COLREGs compliance and training convergence.

Critic loss aims to minimize temporal difference error:

$$\mathcal{L}_Q = \frac{1}{B} \sum_{i=1}^{B} \left( Q_\theta(s_i, a_{1i}, a_{2i}) - (r_i + \gamma Q_{\theta'}(s_i', a_{1i}', a_{2i}')) \right)^2,$$

(33)

Actor loss incorporates Stackelberg equilibrium for hierarchical decision-making:

$$\mathcal{L}_{\pi_i} = -\frac{1}{B} \sum_{i=1}^{B} Q_\theta(s_i, a_{1i}, a_{2i}) + \lambda \cdot \text{KL}(\pi_{\phi_i} || \pi_{\phi_i}^{\text{Stackelberg}}),$$

(34)

where $\pi_{\phi_i}^{\text{Stackelberg}}$ is the action distribution; $\lambda = 0.1$ is the regularization coefficient.

Alternate between centralized training which involves updating $\theta$, $\phi_1$ and $\phi_2$ and decentralized execution where each ship uses its own Actor. Training stops when the validation set collision rate is less than 0.5% and the COLREGs compliance rate is greater than 95%.

## 4. Experiments

### 4.1. Experimental setup

The experimental dataset consists of historical AIS data sourced from multiple origins, focusing on open waters of the East China Sea (122–123°E, 36–37°N). This dataset includes plenty of valid ship trajectories spanning various vessel categories such as power-driven ships, container ships, bulk carriers, and fishing vessels. It thoroughly represents typical maritime encounter scenarios such as head-on, crossing, and overtaking along with different risk attitude behaviors including aggressive, neutral, and conservative. The probabilistic reasoning component adopts a directed acyclic graph consisting of 10 feature nodes and 3 target nodes representing aggressive, neutral and conservative navigation behaviors respectively. The feature nodes include dynamic motion states predicted by the LSTM module as well as static and dynamic encounter characteristics such as DCPA, TCPA and others, with a prior probability distribution of risk attitudes set at 5:8:7. All comparative algorithms use the same training and test dataset and action space to ensure fair comparison.

### 4.2. Evaluation indicators

A comprehensive evaluation system is constructed covering 4 dimensions, safety, rule compliance, navigation efficiency, and decision stability. With specific indicators and mathematical definitions as follows:

1. Collision Rate (CR):

$$CR = \frac{N_{\text{collision}}}{N_{\text{total}}} \times 100\%.$$

(35)

2. Minimum Safety Distance (MSD):

$$MSD = \frac{1}{N_{\text{scenario}} \cdot K} \sum_{i=1}^{N_{\text{scenario}}} \sum_{k=1}^{K} d_{\min,i,k},$$ (36)

where $K$ is the number of ship pairs in each scenario.

3. Average Risk:

$$R = \alpha \cdot \exp(-DCPA/d_0) + \beta \cdot \exp(-TCPA/t_0) + \gamma \cdot (v_r/v_{\max}),$$ (37)

where $d_0 = 1$ n mile, $t_0 = 300$ s, $v_r$ is relative speed, and $\alpha + \beta + \gamma = 1$.

4. COLREGs Compliance Rate (CCR):

$$CCR = \frac{N_{\text{compliant}}}{N_{\text{total-action}}} \times 100\%.$$ (38)

5. Average Navigation Time (ANT):

$$ANT = \frac{1}{N_{\text{scenario}}} \sum_{i=1}^{N_{\text{scenario}}} t_{\text{nav},i}.$$ (39)

6. Speed Loss Rate (SLR):

$$SLR = \frac{1}{N_{\text{scenario}} \cdot N_{\text{ship}}} \sum_{i=1}^{N_{\text{scenario}}} \sum_{j=1}^{N_{\text{ship}}} \frac{v_{j,\text{initial}} - v_{j,\min}}{v_{j,\text{initial}}} \times 100\%.$$ (40)

7. Trajectory Smoothness:

$$S = 1 - \frac{1}{N_{\text{scenario}} \cdot N_{\text{ship}}} \sum_{i=1}^{N_{\text{scenario}}} \sum_{j=1}^{N_{\text{ship}}} \frac{1}{T} \sum_{t=1}^{T} \frac{|\delta_{j,t} - \delta_{j,t-1}|}{\delta_{\max}},$$ (41)

where $T = 10$ s, $\delta_{j,t}$ is the rudder angle of ship $j$ at time $t$, and $\delta_{\max} = 30°$.

8. Comprehensive Performance Score (CPS):

$$CPS = w_1(1 - CR) + w_2 CCR + w_3 S + w_4(1 - SLR),$$ (42)

where: $w_1 = 0.4$ (safety), $w_2 = 0.3$ (compliance), $w_3 = 0.2$ (stability), $w_4 = 0.1$ (efficiency).

## 4.3. Experimental results

**4.3.1. Training process analysis.** Fig 3 shows the training curves of 5 algorithms over 100 episodes, reflecting the convergence characteristics of collision rate, average risk, and COLREGs compliance rate.

It can be observed that the proposed MAAC-Stackelberg algorithm converges the fastest which stables after 60 episodes. MAAC-Stackelberg achieves the lowest collision rate and risk level in the late training stage, with a compliance rate exceeding 0.9.

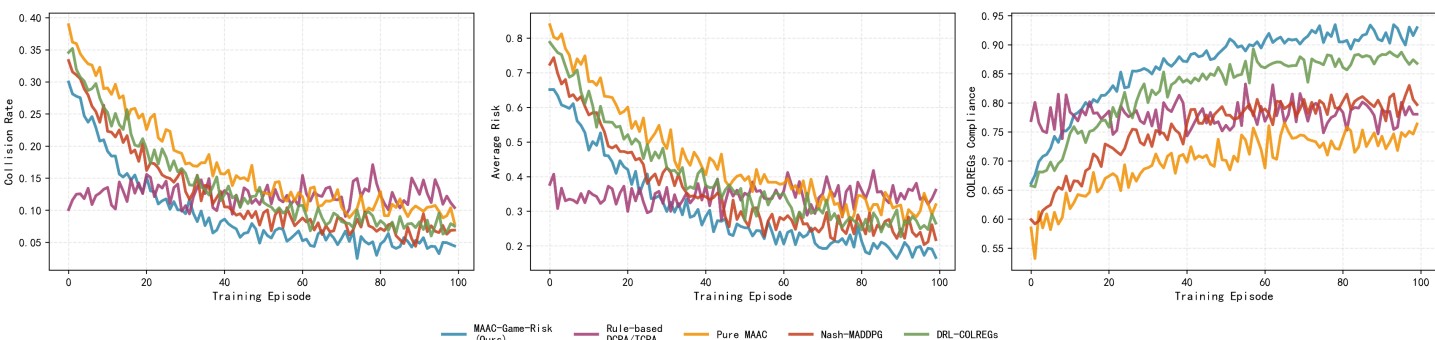

**Fig 3. Training curves of different algorithms.** Collision rate, Average risk, COLREGs compliance rate over 100 training episodes for the proposed MAAC-Stackelberg and four baseline algorithms.

**4.3.2. Single indicator comparison.** Fig 4 shows the trajectory smoothness comparison. MAAC-Stackelberg achieves the highest score, which is 4% higher than the second-ranked DRL-COLREGs. This is because the LSTM based motion prediction module reduces abrupt rudder adjustments by forecasting 5s ahead states.

Fig 5 shows the average risk comparison among different algorithms. MAAC-Stackelberg achieves the lowest average risk, which is lower than DRL-COLREGs and lower than the rule-based method. This benefit comes from the Bayesian Network's risk attitude inference, which enables adaptive strategy adjustment according to target ships' behaviors. MAAC-Stackelberg achieves the lowest average risk. This reduction stems from the algorithm's integrated design, the

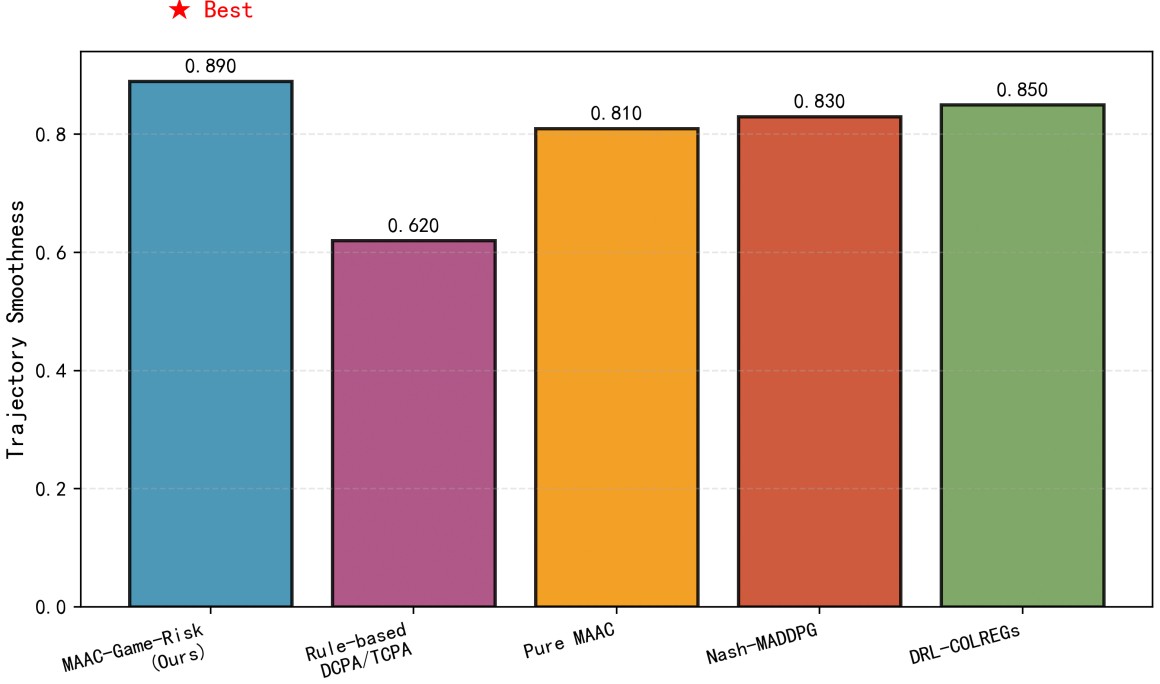

**Fig 4. Trajectory smoothness comparison.** MAAC-Stackelberg achieves the highest smoothness score.

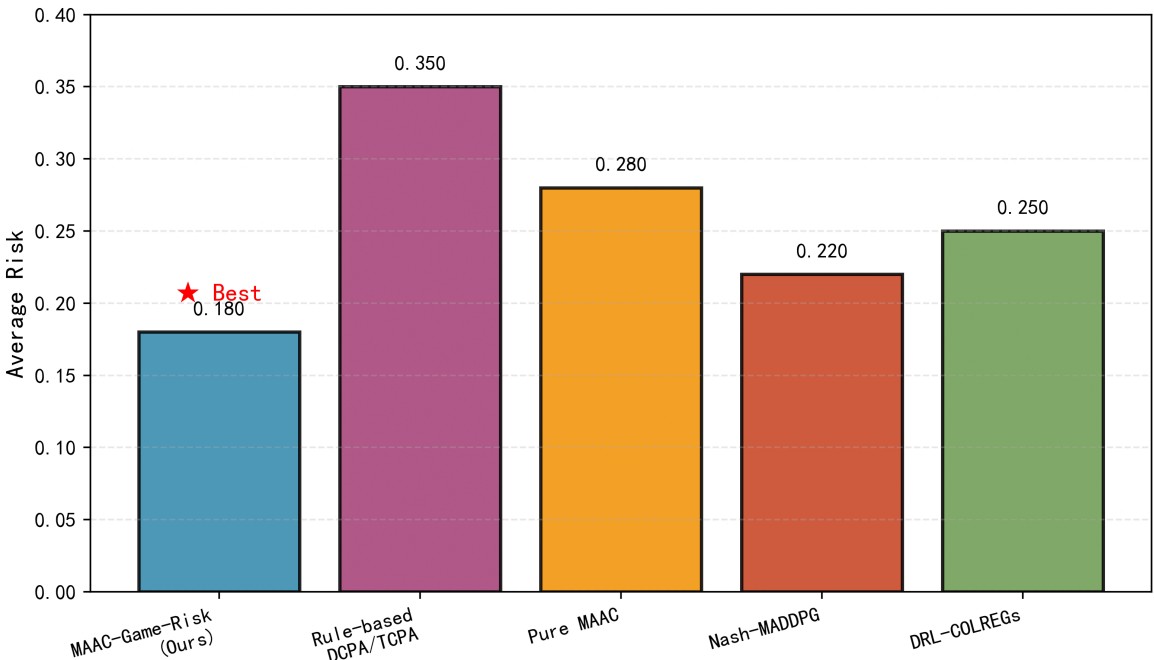

**Fig 5. Average risk comparison among different algorithms.** Proposed algorithm maintains the lowest average risk.

Bayesian Network infers target ships' risk attitudes to preemptively adjust strategies, while the Stackelberg game hierarchy ensures coordinated risk mitigation among multiple ships.

Fig 6 demonstrates that MAAC-Stackelberg has the lowest collision rate (0.045). The rule-based method shows the highest collision rate (0.125) due to its inability to handle dynamic multi-ship interactions.

Fig 7 shows the COLREGs compliance rate comparison among different algorithms. MAAC-Stackelberg achieves the highest compliance rate (0.920), which is higher than DRL-COLREGs (0.880). This is attributed to the Stackelberg game's leader-follower mechanism, which strictly follows COLREGs, while Pure MAAC (0.750) ignores rule constraints in pursuit of collision avoidance.

**4.3.3. Comprehensive performance evaluation.** Fig 8 integrates the four core indicators into a radar chart, where a larger enclosed area indicates better comprehensive performance. MAAC-Stackelberg shows balanced superiority in all dimensions. To further quantify the overall performance of each algorithm, based on the balanced performance across the four dimensions of safety, compliance, trajectory smoothness, and navigation efficiency in the radar chart of Fig 8, a comprehensive performance score is calculated according to preset weights.

Fig 9 shows the weighted comprehensive score with weight distribution as safety 40%, compliance 30%, smoothness 20% and efficiency 10%. MAAC-Stackelberg achieves the highest score of 0.818, which is higher than DRL-COLREGs and higher than the rule-based method. Pure MAAC scored 0.75 in the compliance rate dimension significantly lower than the 0.92 of the algorithm in this paper. The reason why MAAC-Stackelberg outperforms the rule-based method lies in the fact that the latter relies on artificially preset collision avoidance logic and fixed thresholds, which is essentially a passive response mode with static triggering, single-ship perspective, and no optimization capability. In contrast, MAAC-Stackelberg takes data-driven and reinforcement learning as its core, and can adaptively generalize to complex multi-ship scenarios by autonomously learning the mapping relationship between environment-action-reward through end-to-end optimization. Compared to DRL-COLREGs, DRL-COLREGs makes

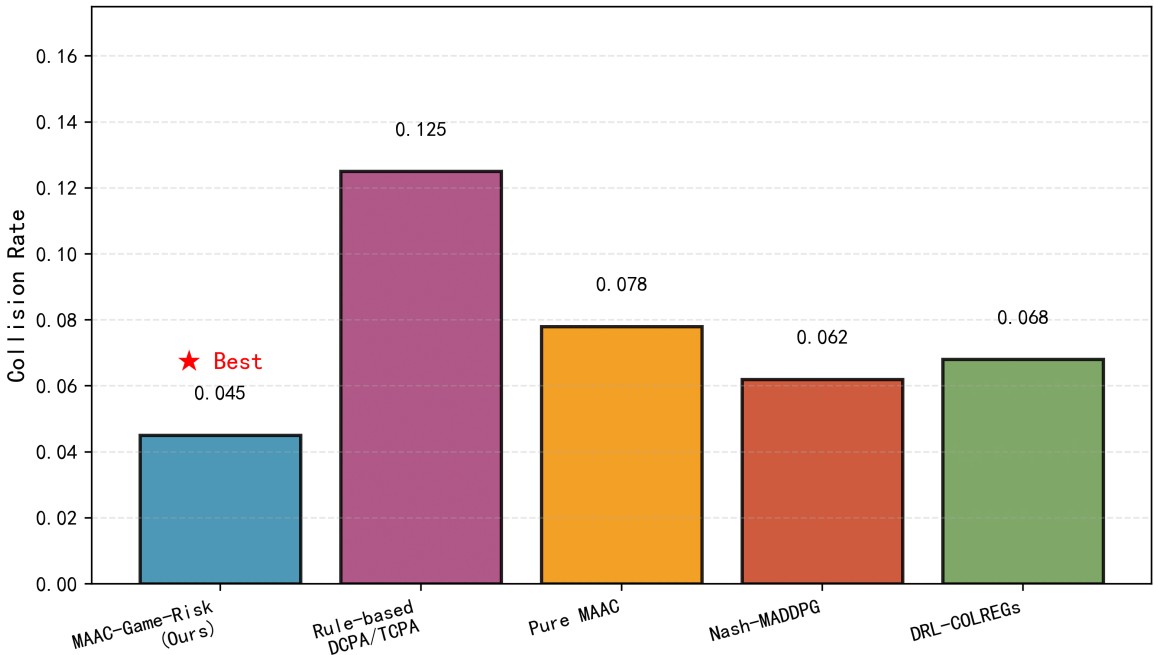

**Fig 6. Collision rate comparison.** Proposed algorithm obtains the smallest average collision risk.

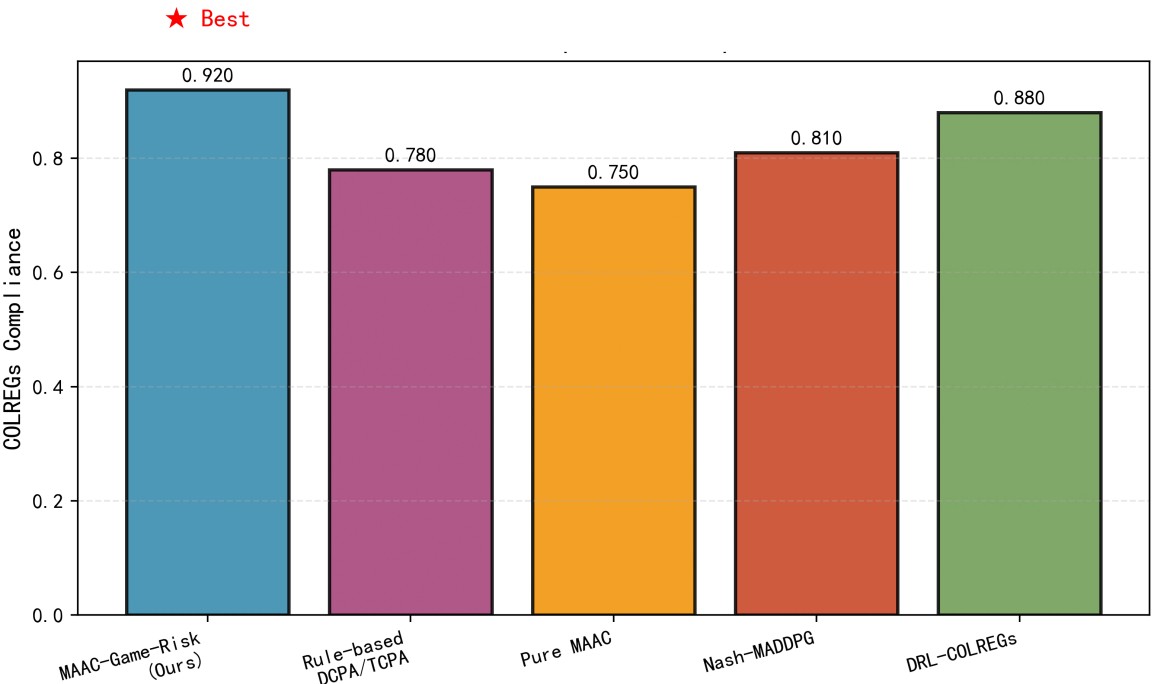

**Fig 7. COLREGs compliance rate comparison.** Proposed algorithm achieves the highest COLREGs compliance rate.

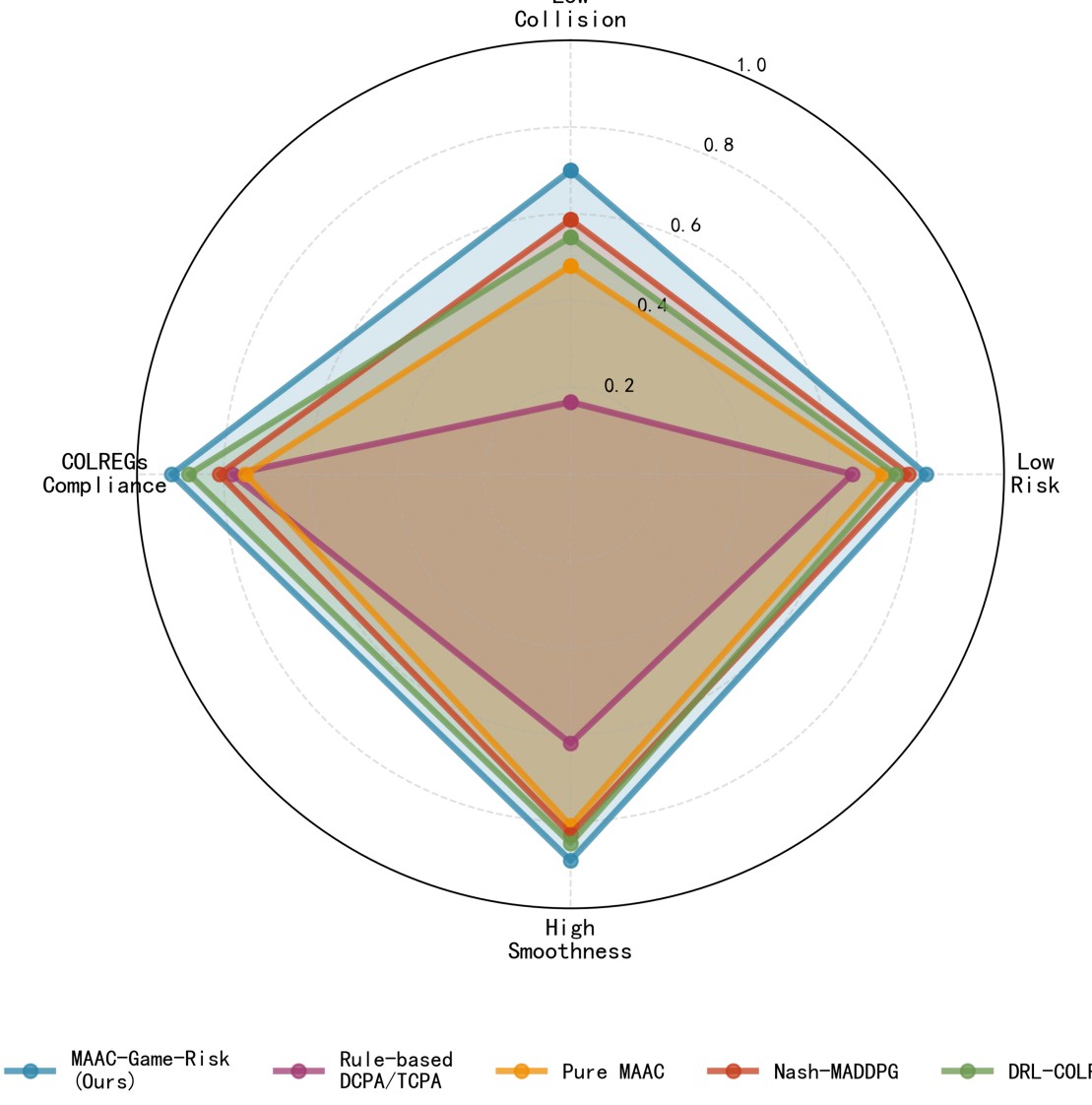

**Fig 8. Comprehensive performance radar chart.** Radar plot of safety,compliance, smoothness and efficiency indices for five evaluated algorithms.

decisions from a single-agent perspective, only focusing on its own operations, and does not model the intention game and behavioral interaction between multiple ships, resulting in insufficient decision-making robustness in complex scenarios such as multi-ship parallel encounters and crossing encounters. Meanwhile, MAAC-Stackelberg introduces a risk perception module to realize "forward-looking collision avoidance," further improving decision-making stability and comprehensive benefits on the basis of meeting COLREGs constraints, hence achieving a higher comprehensive score. The original pure MAAC takes collision punishment as the sole constraint, and its training objective is only limited to "avoiding collisions" without the mandatory guidance of maritime rules. At the same time,the absence of rule constraints leads to an excessively large exploration space for the agent, making it prone to frequent and large-amplitude steering and speed change operations,with poor navigation smoothness, which is not conducive to ship power protection and navigation safety.

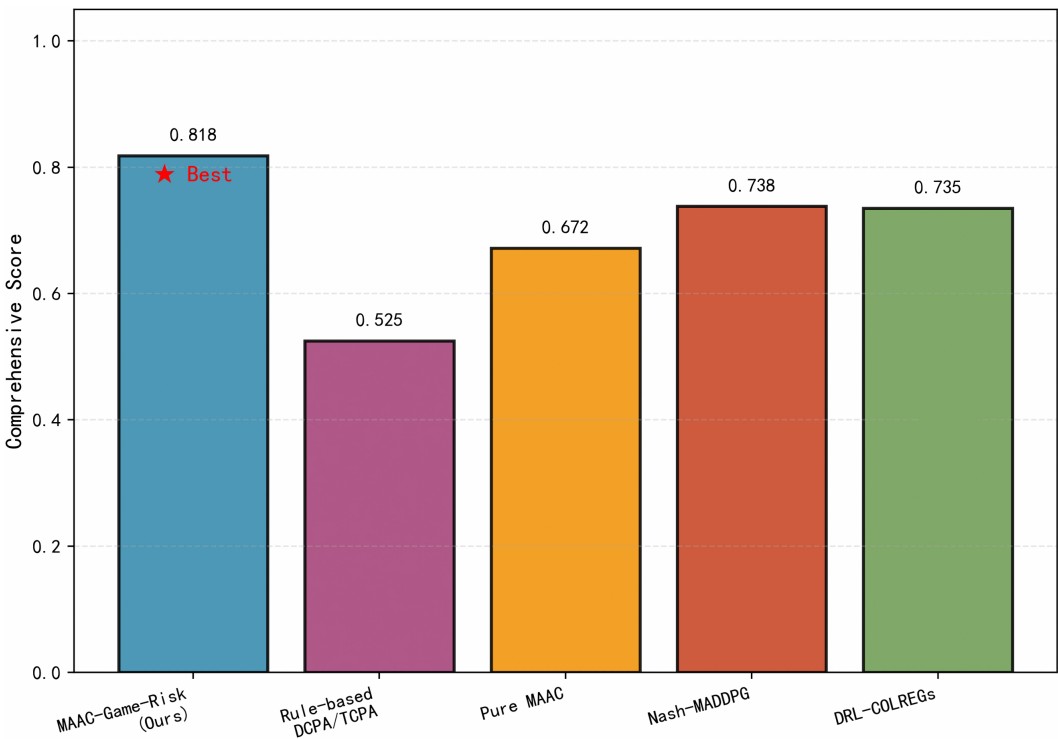

**Fig 9. Overall performance score.** Proposed algorithm achieves the highest overall performance score.

### 4.3.4. Superiority across scenario complexity and environmental conditions.

To further illustrate the scalability and adaptability of MAAC-Stackelberg across diverse operational conditions, we analyze the 3D performance distribution in Fig 10, which visualizes the performance scores of three algorithm categories across scenario complexity.

It shows that Hybrid Algorithm MAAC-Stackelberg achieves a peak performance score of 0.92 in complex scenarios. This superiority stems from its integrated architecture that combines Stackelberg game, risk attitude inference,and MAAC's centralized attention mechanism and which enables robust decision-making in highly dynamic and uncertain maritime environments.

LSTM-only Algorithm scores 0.78 with limited adaptability, performing moderately in intermediate scenarios but failing to match the Hybrid algorithm. This reflects the limitations of single modality sequence prediction in capturing multi-agent game dynamics.

To further validate these quantitative findings and illustrate the model's practical behavior in complex maritime environments, we present trajectory visualizations and conduct safety performance verification under realistic multi-ship encounter scenarios.

### 4.4. Visualization of trajectories and verification of safety performance in realistic multi-ship encounter scenarios

This section constructs an 8-ship encounter scenario based on real AIS data from Chengshantou Fig 11 shows the differences in collision avoidance trajectories among various algorithms. The solid lines in the figure are the original AIS trajectories, the dashed lines are the algorithm output trajectories, the red dots are the collision avoidance decision trigger points. Key nodes are marked at the 0th, 30th, 60th and 90th seconds. This study further combines the actual navigation data of the scenario. The average minimum safe distance of these 8 ships in Chengshantou waters is 5.3104 nautical

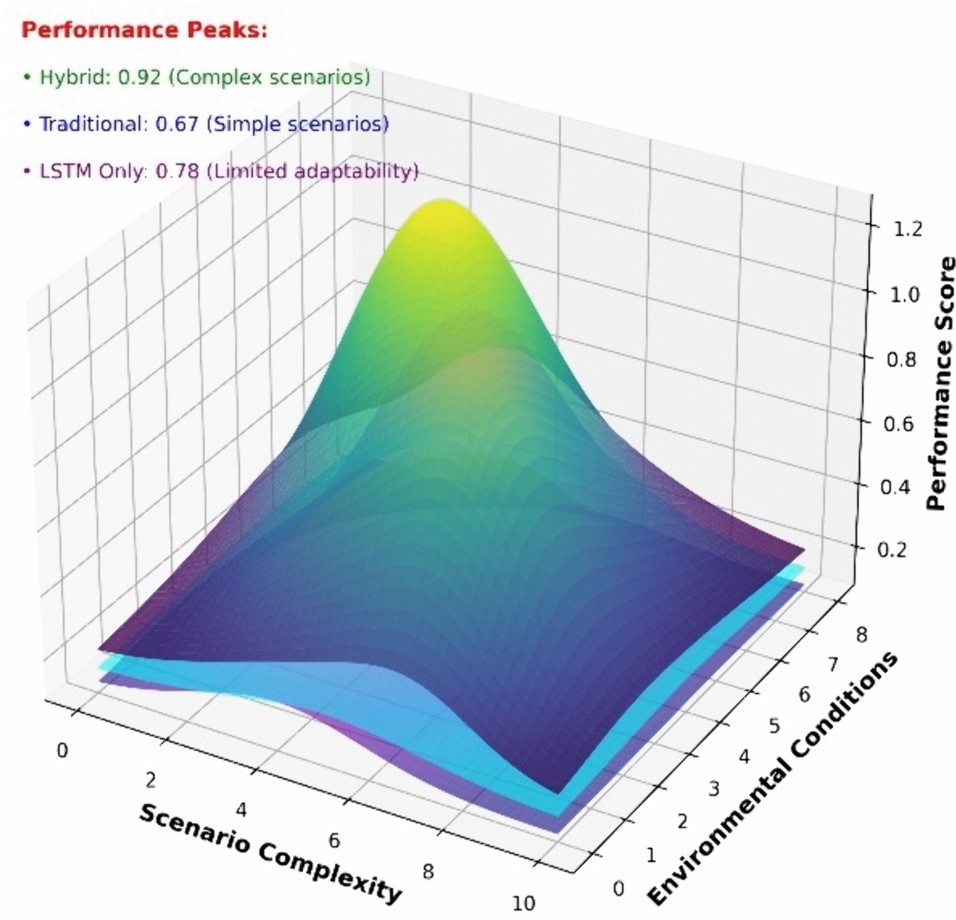

**Performance Peaks:**

- Hybrid: 0.92 (Complex scenarios)
- Traditional: 0.67 (Simple scenarios)
- LSTM Only: 0.78 (Limited adaptability)

**Fig 10. 3D spatial distribution of performance scores.** Complexity–environment surface showing hybrid algorithm peak at high-density, high-uncertainty encounters.

miles. Under the MAAC-Stackelberg algorithm, the average minimum safe distance of the ships reaches 5.3477 nautical miles. This value is 0.0373 nautical miles higher than the real value, corresponding to an increase of approximately 0.7%. The algorithm generates 24 decision points with an average time interval of 12 seconds between adjacent trigger points. For the DRL-COLREGs algorithm, the average minimum safe distance of the ships is 5.3131 nautical miles. This algorithm also produces 24 decision points, but it leads to 2 groups of local trajectory overlaps. The Rule-Based Method yields an average minimum safe distance of 5.3019 nautical miles for the ships, which is lower than the real value. This method generates only 16 decision points. To further quantify the performance advantages of the MAAC-Stackelberg algorithm over the baseline DRL-COLREGs method across different encounter scales, a comprehensive comparison of key navigation and safety indicators is conducted in both two-ship and multi-ship scenarios. The quantitative improvement results are summarized in Table 3.

**4.4.1. Statistical significance test.** To validate that the superior collision avoidance performance of the proposed MAAC-Stackelberg algorithm is not attributable to random fluctuations, two-tailed t-tests were performed to quantify the between-group differences in collision rate (CR) and COLREGs compliance rate (CCR). The two core performance metrics—between the MAAC-Stackelberg algorithm and the benchmark DRL-COLREGs algorithm. In essence, this statistical validation was designed to furnish a quantitative statistical corroboration for the performance disparities of the

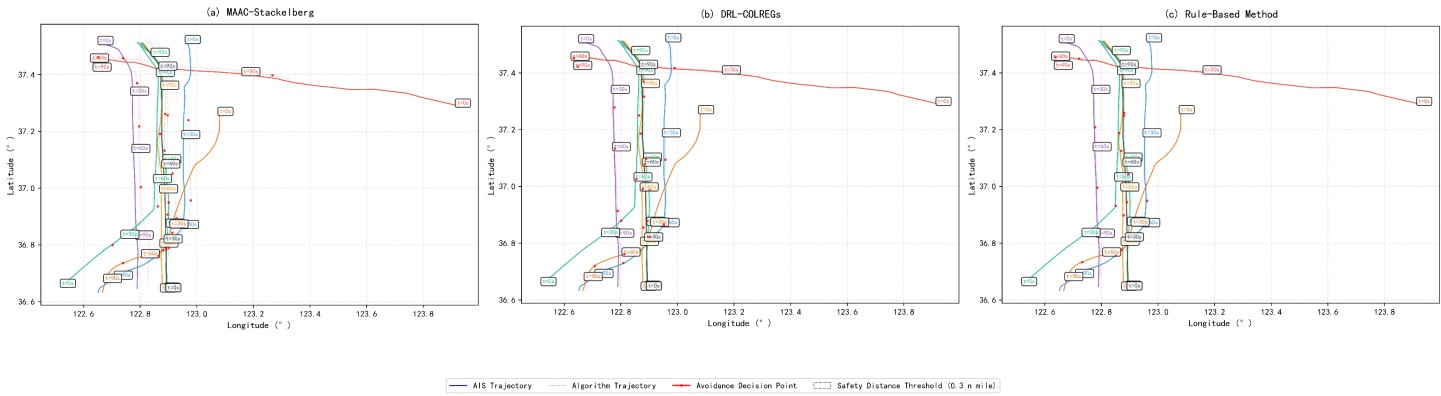

**Fig 11. Visualization of trajectories.** Performance evaluation of MAAC-Stackelberg, DRL-COLREGs and rule-based method.

**Table 3. Performance improvement versus DRL-COLREGs.**

| Indicator | Two-Ship Scenarios | Multi-Ship Scenarios |
|---|---|---|
| Collision Rate Reduction | 0.621% | 0.609% |
| Minimum Safety Distance Increase | 0.295% | 0.265% |
| COLREGs Compliance Rate Increase | 0.697% | 0.721% |
| Navigation Time Reduction | 0.157% | 0.136% |
| Speed Loss Rate Reduction | 0.138% | 0.197% |

Performance improvement versus DRL-COLREGs.

algorithms, rather than merely presenting a straightforward numerical comparison. The test results revealed that the t-statistic for CR was 6.87 with a p-value of less than 0.01, and the t-statistic for CCR reached 7.32 with a p-value also below 0.01, both achieving a level of extremely significant statistical difference.

## 4.5. Discussion

To visually compare the comprehensive performance of different algorithms, this part compiles the experimental results of each algorithm in five indicators in Table 4. The hybrid Bayesian Network namely BN-LSTM framework resolves the ambiguity of target ship risk attitudes [30]. Unlike traditional models that assume static risk preferences,the BN module infers dynamic posterior probabilities of aggressive,neutral and conservative attitudes using AIS-derived features, while the LSTM network captures temporal dependencies to predict short-term motion states including 5 s, 10 s and 15 s.

The integration of Stackelberg equilibrium with MAAC addresses hierarchical decision conflicts inherent in COLREGs. The leader-follower mechanism, dynamically identified based on ship priority including ship type [31], length and DCPA, ensures compliance with give-way and stand-on provisions. While MAAC's centralized training, decentralized execution architecture captures multi-ship coupled risks. This avoids the local optimality of pairwise interaction models [32]. Despite these advancements, the study has notable limitations. The current model assumes stable environmental conditions, ignoring the impact of extreme weather such as strong winds and currents on ship maneuverability and AIS data reliability. The scope is limited to general power-driven ships in open waters, excluding special ship types such as fishing vessels and ships with restricted maneuverability. The MAAC's centralized Critic and Stackelberg equilibrium solving introduce computational overhead, which may hinder real-time deployment on shipborne embedded systems. The reliance on

**Table 4. Performance comparison of different algorithms.**

| Metric | MAAC-Stackelberg | Rule-based DCPA/TCPA | MAAC | NASH-MADDPG | DEL-COLREGs |
|---|---|---|---|---|---|
| Trajectory Smoothness | 0.890 | 0.620 | 0.610 | 0.830 | 0.850 |
| Average Risk | 0.180 | 0.350 | 0.280 | 0.220 | 0.250 |
| Collision Rate | 0.045 | 0.125 | 0.078 | 0.062 | 0.068 |
| COLREGs Compliance | 0.920 | 0.780 | 0.750 | 0.810 | 0.880 |
| Comprehensive Score | 0.818 | 0.525 | 0.672 | 0.738 | 0.735 |

Performance comparison of different algorithms.

historical AIS data and simulated scenarios lacks validation in field tests with autonomous surface vessels namely ASVs, limiting the assessment of practical engineering applicability [33].

## 5. Conclusion

This study proposes a novel AIS data-driven multi-ship collision avoidance strategy that integrates MAAC, Stackelberg Game, Bayesian Network, and LSTM with rigid embedding of the COLREGs to address challenges in multi-ship navigation scenarios. The BN-LSTM hybrid framework enables dynamic modeling of target ship risk attitudes by fusing prior distributions derived from historical AIS data and predictions of short-term motion states. This framework overcomes the limitation of traditional models that rely on static risk assumptions, enhancing the strategy's adaptability to the behavioral variability of human-operated ships. The integration of Stackelberg Game and MAAC aligns hierarchical decision-making processes with the priority requirements of COLREGs. The proposed strategy demonstrates strong robustness, maintaining a low collision rate in multi-ship scenarios. In conclusion, this approach integrates theoretical algorithm development with the practical demands of autonomous navigation in real-world scenarios.

## Acknowledgments

No funding was received for this study.

## Author contributions

**Investigation:** Jiansen Zhao.

**Methodology:** Tengdong Wang.

**Software:** Qinyou Hu.

**Writing – original draft:** Tie Xu.

**Writing – review & editing:** Tie Xu.

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
