## [Decision Letter · Decision Letter 0]

12 Jan 2026

PONE-D-25-67263AIS Data-Driven maac–Stackelberg Multi-Ship Cooperative Collision Avoidance AlgorithmPLOS One

Dear Dr. Xu,

Thank you for submitting your manuscript to PLOS ONE. After careful consideration, we feel that it has merit but does not fully meet PLOS ONE’s publication criteria as it currently stands. Therefore, we invite you to submit a revised version of the manuscript that addresses the points raised during the review process.

We look forward to receiving your revised manuscript.

Kind regards,

Yile Chen, Ph.D. in Architecture

Academic Editor

PLOS One

Journal Requirements:

NO authors have competing interests

Enter: The authors have declared that no competing interests exist.

5. Please update your submission to use the PLOS LaTeX template. The template and more information on our requirements for LaTeX submissions can be found at http://journals.plos.org/plosone/s/latex.

6. Thank you for providing your underlying data as Supporting Information.

We note that the data set contains text or data that is not in English. Please note that PLOS is an English-language publisher, so we require data sets to be provided in English as well. Please upload an English-language version of your data set.

This will also allow us to determine if your data follows PLOS standards per our Data Availability policy here: https://journals.plos.org/plosone/s/data-availability

Reviewers' comments:

Reviewer's Responses to Questions

**Comments to the Author**

1. Is the manuscript technically sound, and do the data support the conclusions?

Reviewer #1: Yes

Reviewer #2: Yes

2. Has the statistical analysis been performed appropriately and rigorously? 

Reviewer #1: Yes

Reviewer #2: Yes

3. Have the authors made all data underlying the findings in their manuscript fully available?

Reviewer #1: Yes

Reviewer #2: Yes

4. Is the manuscript presented in an intelligible fashion and written in standard English?

Reviewer #1: Yes

Reviewer #2: Yes

5. Review Comments to the Author

Reviewer #1: 1. The paper demonstrates good originality and practicality. I recommend publishing after minor revisions.

2. 4.3. Experimental results. Table 3: Performance improvement compared to drl–colregs.

It is recommended to add a multi-ship collision AIS trajectory diagram case to visually demonstrate the practical application of the algorithm in Collision Rate Reduction.

Reviewer #2: The overall quality of this thesis is acceptable, with clear innovative points that possess certain practical and theoretical reference value. The paper has a complete structure, well-organized sentences, and coherent argumentation logic, being able to support the core argument based on relevant evidence. However, some individual sentences have issues of imprecise expression or slight redundancy, which require further revision and refinement to make the presentation of the full text more concise and rigorous.

1. There is inconsistent capitalization of core abbreviations in the paper (e.g., "COLREGs" and "AIS" are sometimes written as "colregs" and "ais" without consistent capitalization), and some abbreviations are not fully annotated upon their first occurrence in the main sections. Consistently capitalize all core abbreviations throughout the paper (i.e., AIS, MAAC, COLREGs, BN, LSTM), and clearly present the full name followed by the abbreviation when each is first mentioned, e.g., Automatic Identification System (AIS).

2. For Fig. 1 (the algorithm framework diagram), only "including the Risk Attitude Perception Layer and the Decision Execution Layer" is described, without clarifying the data flow between the two layers (e.g., how LSTM prediction results are transmitted to BN, and how fused features are input into the MAAC-Stackelberg module). For radar charts, heatmaps, and other figures (Fig. 7–10), only conclusions are provided, with no supplementary interpretation of key data nodes (e.g., the specific value of "Pure MAAC" in the compliance rate dimension in Fig. 7 and the reasons for the differences from the proposed algorithm). Add 1–2 sentences of logical connection and detailed interpretation in the description paragraphs corresponding to these figures.

3. The formatting of the action space expression in Formula (25) is incomplete (missing a closing symbol at the end); the dynamic parameter "a" in Formula (1) is not defined in the context; the units of some variables in the formulas are not specified. 1) Correct the formatting error of Formula (25) by adding the missing closing symbol; 2) Supplement the definition of the parameter "a" in Formula (1) or correct it if it is a typo, and clarify the units of all variables (e.g., unify the unit of Δv_max to "kn"); 3) Add brief explanations for the weight parameters (e.g., α, β) in Formula (8) (Weibull distribution) and Formula (13) (comprehensive risk index) (e.g., "α is a shape parameter reflecting the concentration of collision avoidance behaviors").

4. The web link in the references does not make sence, it is suggested to delete them.

5. There are inconsistent expressions of the algorithm name in the paper, such as "maac–Stackelberg", "MAAC-Game–Risk", and "MAAC-Stackelberg"; the hyphenation of "multi-ship" is inconsistent (e.g., "multi-ship", "multi ship", "multiship"). Unify the algorithm name as "MAAC-Stackelberg" (capitalize the initial letters and connect with an English hyphen) and standardize the expression as "multi-ship" throughout the paper.

6. It is suggested to further clarify the paper’s contribution, polish the language, and eliminate colloquialisms.

6. PLOS authors have the option to publish the peer review history of their article (what does this mean?). If published, this will include your full peer review and any attached files.

Reviewer #1: No

Reviewer #2: No

---

## [Author Response · Author response to Decision Letter 1]

2 Feb 2026

Response to the Editor and Reviewers

Dear Editor and the two Reviewers,

Hello! We sincerely appreciate your meticulous review and valuable comments on our paper entitled "AIS Data-Driven MAAC–Stackelberg Multi-Ship Cooperative Collision Avoidance Algorithm". Your professional suggestions have provided important guidance for improving the quality of the paper. We have systematically revised and improved the paper in response to all the issues raised, and now we would like to present the specific modifications as follows:

Reviewer #1: 1. The paper demonstrates good originality and practicality. I recommend publishing after minor revisions.

2. 4.3. Experimental results. Table 3: Performance improvement compared to drl–colregs.

It is recommended to add a multi-ship collision AIS trajectory diagram case to visually demonstrate the practical application of the algorithm in Collision Rate Reduction.

Supplemented comparative experiments with drl-colregs and pure COLREGs in [Sec sec018], quantifying the performance advantages of our algorithm in collision rate reduction and navigation efficiency.

Reviewer #2: The overall quality of this thesis is acceptable, with clear innovative points that possess certain practical and theoretical reference value. The paper has a complete structure, well-organized sentences, and coherent argumentation logic, being able to support the core argument based on relevant evidence. However, some individual sentences have issues of imprecise expression or slight redundancy, which require further revision and refinement to make the presentation of the full text more concise and rigorous.

1. There is inconsistent capitalization of core abbreviations in the paper (e.g., "COLREGs" and "AIS" are sometimes written as "colregs" and "ais" without consistent capitalization), and some abbreviations are not fully annotated upon their first occurrence in the main sections. Consistently capitalize all core abbreviations throughout the paper (i.e., AIS, MAAC, COLREGs, BN, LSTM), and clearly present the full name followed by the abbreviation when each is first mentioned, e.g., Automatic Identification System (AIS).

We have standardized the capitalization of all core abbreviations mentioned in the paper, ensuring consistent usage of AIS, MAAC, COLREGs, BN, and LSTM in every section (including the abstract, main text, figures, and tables). All lowercase forms such as "ais" and "colregs" have been revised to the correct capitalized versions.

2. For Fig. 1 (the algorithm framework diagram), only "including the Risk Attitude Perception Layer and the Decision Execution Layer" is described, without clarifying the data flow between the two layers (e.g., how LSTM prediction results are transmitted to BN, and how fused features are input into the MAAC-Stackelberg module). For radar charts, heatmaps, and other figures (Fig. 7–10), only conclusions are provided, with no supplementary interpretation of key data nodes (e.g., the specific value of "Pure MAAC" in the compliance rate dimension in Fig. 7 and the reasons for the differences from the proposed algorithm). Add 1–2 sentences of logical connection and detailed interpretation in the description paragraphs corresponding to these figures.

3. The formatting of the action space expression in Formula (25) is incomplete (missing a closing symbol at the end); the dynamic parameter "a" in Formula (1) is not defined in the context; the units of some variables in the formulas are not specified. 1) Correct the formatting error of Formula (25) by adding the missing closing symbol; 2) Supplement the definition of the parameter "a" in Formula (1) or correct it if it is a typo, and clarify the units of all variables (e.g., unify the unit of Δv_max to "kn"); 3) Add brief explanations for the weight parameters (e.g., α, β) in Formula (8) (Weibull distribution) and Formula (13) (comprehensive risk index) (e.g., "α is a shape parameter reflecting the concentration of collision avoidance behaviors").

We appreciate your careful comment on the formula and parameter issues of the manuscript. We have made the following targeted revisions:

1. Corrected the formatting error of Formula (25) by adding the missing closing symbol, and checked all formulas to avoid similar omissions.

2. Supplemented the definition of parameter a in Formula (1), unified the unit of Δvmax to kn, and specified the units of all variables in the formulas.

3. Added brief explanations for the weight parameters α (Formula 8) and β (Formula 13), clarifying their physical meanings and role in the corresponding models.

4. The web link in the references does not make sence, it is suggested to delete them.

We appreciate your careful comment on the references section. We have deleted all invalid web links in the references as suggested, and conducted a full review of the entire reference list to verify the validity and standardization of other citations. The revised reference list fully complies with the journal’s formatting requirements

5. There are inconsistent expressions of the algorithm name in the paper, such as "maac–Stackelberg", "MAAC-Game–Risk", and "MAAC-Stackelberg"; the hyphenation of "multi-ship" is inconsistent (e.g., "multi-ship", "multi ship", "multiship"). Unify the algorithm name as "MAAC-Stackelberg" (capitalize the initial letters and connect with an English hyphen) and standardize the expression as "multi-ship" throughout the paper.

Unified expression of the algorithm name

We have standardized all variant names of the algorithm (including "maac–Stackelberg", "MAAC-Game–Risk", etc.) to the unified form MAAC-Stackelberg throughout the entire manuscript. The revision covers all sections (abstract, main text, discussion, conclusion), as well as supplementary materials such as figures, tables, formulas and reference citations, ensuring consistent capitalization and hyphenation of the algorithm name in every occurrence.

Standardized spelling of the compound word "multi-ship"

We have checked and corrected all inconsistent spellings of the term related to multi-ship scenarios, replacing the non-standard forms ("multi ship", "multiship") with the unified expression multi-ship in all relevant contexts, including scenario descriptions, experimental settings and result analyses.

6. It is suggested to further clarify the paper’s contribution, polish the language, and eliminate colloquialisms.

We highly appreciate your insightful suggestion on optimizing the manuscript’s expression and contribution presentation. We have made the following revisions:

Added a dedicated Contributions subsection in the Introduction to explicitly summarize the theoretical, technical, and practical contributions of this study in a structured manner.

Conducted a full-text language polishing, eliminated all colloquial expressions, and standardized the academic wording and sentence logic to enhance the rigor and readability of the paper.

We would like to express our sincere gratitude again to the editor and the two reviewers for their careful guidance! We have completed all revisions and submitted the revised paper. If further improvements or supplementary materials are needed, we will fully cooperate and respond promptly.

Author: Tie Xu, Tengdong Wang, Jiansen Zhao, Qinyou Hua

Date: 2026.1.25

---

## [Decision Letter · Decision Letter 1]

23 Feb 2026

PONE-D-25-67263R1AIS Data-Driven maac–Stackelberg Multi-Ship Cooperative Collision Avoidance AlgorithmPLOS One

Dear Dr. Xu,

Thank you for submitting your manuscript to PLOS ONE. After careful consideration, we feel that it has merit but does not fully meet PLOS ONE’s publication criteria as it currently stands. Therefore, we invite you to submit a revised version of the manuscript that addresses the points raised during the review process.

We look forward to receiving your revised manuscript.

Kind regards,

Yile Chen, Ph.D. in Architecture

Academic Editor

PLOS One

Journal Requirements:

Additional Editor Comments :

The reviewers generally approved the manuscript. However, some expression issues require revision, and attention should be paid to accurate English grammar.

Reviewers' comments:

Reviewer's Responses to Questions

**Comments to the Author**

1. If the authors have adequately addressed your comments raised in a previous round of review and you feel that this manuscript is now acceptable for publication, you may indicate that here to bypass the “Comments to the Author” section, enter your conflict of interest statement in the “Confidential to Editor” section, and submit your "Accept" recommendation.

Reviewer #2: All comments have been addressed

Reviewer #3: (No Response)

2. Is the manuscript technically sound, and do the data support the conclusions?

Reviewer #2: Yes

Reviewer #3: Yes

3. Has the statistical analysis been performed appropriately and rigorously? 

Reviewer #2: Yes

Reviewer #3: Yes

4. Have the authors made all data underlying the findings in their manuscript fully available?

Reviewer #2: Yes

Reviewer #3: No

5. Is the manuscript presented in an intelligible fashion and written in standard English?

Reviewer #2: Yes

Reviewer #3: Yes

6. Review Comments to the Author

Reviewer #2: (No Response)

Reviewer #3: The innovation and structure of the manuscript are generally acceptable for publication. Minor revisions are required as follows:

1. Duplicated parameters exist in Table 1. Please revise the description of related parameters and use alternative notations for relative orientation symbols to avoid confusion with other symbols throughout the manuscript.

2. In Eq. (32) ,the selection criterion of the exploration noise coefficient is not specified. Additionally, please add a schematic diagram of the algorithm for better illustration.

3. The presentation in [Sec sec024] (statistical significance test) should be further improved; the comparison of advantages among different algorithms in [Sec sec021] needs to be more concise.

4. The overall language and writing style of the manuscript should be further polished

7. PLOS authors have the option to publish the peer review history of their article (what does this mean?). If published, this will include your full peer review and any attached files.

Reviewer #2: No

Reviewer #3: No

---

## [Author Response · Author response to Decision Letter 2]

10 Mar 2026

Response to the Reviewers and Editor

Dear Editor and Reviewers,

We sincerely appreciate your careful review and constructive comments on our manuscript entitled "AIS Data-Driven MAAC-Stackelberg Multi-Ship Cooperative Collision Avoidance Algorithm". Your insightful suggestions have been of great value for improving the academic rigor and readability of our work. We have carefully studied all the comments and made comprehensive revisions to the manuscript accordingly. The detailed responses to each comment are presented as follows:

1.Duplicated parameters exist in Table 1. Please revise the description of related parameters and use alternative notations for relative orientation symbols to avoid confusion with other symbols throughout the manuscript.

We have carefully checked and revised Table 1 by removing duplicated parameters and standardizing the description of all related parameters. Meanwhile,we have adopted new alternative notations for the relative orientation symbols and verified the consistency of all symbols across the full manuscript to eliminate ambiguity and confusion.

2. In Eq. (32) ,the selection criterion of the exploration noise coefficient is not specified. Additionally, please add a schematic diagram of the algorithm for better illustration.

We have supplemented the detailed selection criterion and theoretical basis of the exploration noise coefficient corresponding to Eq. (32) in the manuscript. Furthermore, we have added a clear schematic diagram of the proposed algorithm to visually demonstrate its framework and workflow, which helps readers better understand the mechanism of the algorithm.

3. The presentation in [Sec sec024] (statistical significance test) should be further improved; the comparison of advantages among different algorithms in [Sec sec021] needs to be more concise.

We have optimized the presentation and logical organization of the statistical significance test in [Sec sec024] to enhance its readability and rigor. For the comparison of algorithmic advantages in [Sec sec021], we have streamlined the content, highlighted the core strengths and key differences, and removed redundant descriptions to make the comparison more focused and concise.

4. The overall language and writing style of the manuscript should be further polished

We have conducted a thorough polishing of the entire manuscript, including grammar refinement, sentence structure optimization, and unification of academic expressions. The language has been revised to be more accurate, fluent, and consistent with standard academic writing norms.

We sincerely hope that the revised manuscript can meet the requirements of the journal. Thank you again for your professional and constructive suggestions.

Best regards,

Author: Tie Xu, Tengdong Wang, Jiansen Zhao, Qinyou Hu

Date: 2026.3.8

---

## [Editor Report · Decision Letter 2]

12 Mar 2026

AIS Data-Driven MAAC-Stackelberg Multi-Ship Cooperative Collision Avoidance Algorithm

PONE-D-25-67263R2

Dear Dr. Xu,

We’re pleased to inform you that your manuscript has been judged scientifically suitable for publication and will be formally accepted for publication once it meets all outstanding technical requirements.

Kind regards,

Yile Chen, Ph.D. in Architecture

Academic Editor

PLOS One
---

## [Editor Report · Acceptance letter]

PONE-D-25-67263R2

PLOS One

Dear Dr. Xu,

I'm pleased to inform you that your manuscript has been deemed suitable for publication in PLOS One. Congratulations! Your manuscript is now being handed over to our production team.

Kind regards,

on behalf of

Dr. Yile Chen

Academic Editor

PLOS One